# Semialgebraic Optimization for Lipschitz Constants of ReLU Networks

**Tong Chen**
LAAS-CNRS
Université de Toulouse
31400 Toulouse, France
tchen@laas.fr

**Jean-Bernard Lasserre**
LAAS-CNRS & IMT
Université de Toulouse
31400 Toulouse, France
lasserre@laas.fr

**Victor Magron**
LAAS-CNRS
Université de Toulouse
31400 Toulouse, France
vmagron@laas.fr

**Edouard Pauwels**
IRIT & IMT
Université de Toulouse
31400 Toulouse, France
edouard.pauwels@irit.fr

## Abstract

The Lipschitz constant of a network plays an important role in many applications of deep learning, such as robustness certification and Wasserstein Generative Adversarial Network. We introduce a semidefinite programming hierarchy to estimate the global and local Lipschitz constant of a multiple layer deep neural network. The novelty is to combine a polynomial lifting for ReLU functions derivatives with a weak generalization of Putinar's positivity certificate. This idea could also apply to other, nearly sparse, polynomial optimization problems in machine learning. We empirically demonstrate that our method provides a trade-off with respect to state of the art linear programming approach, and in some cases we obtain better bounds in less time.

## 1 Introduction

We focus on the multiple layer networks with ReLU activations. We propose a computationally efficient method to give a valid upper bound on the Lipschitz constant of such networks. Recall that a function $f$, defined on a convex set $\mathcal{X} \subseteq \mathbb{R}^n$, is $L$-Lipschitz with respect to the norm $|| \cdot ||$ if for all $\mathbf{x}, \mathbf{y} \in \mathcal{X}$, we have $|f(\mathbf{x}) - f(\mathbf{y})| \leq L||\mathbf{x} - \mathbf{y}||$. The Lipschitz constant of $f$ with respect to norm $|| \cdot ||$, denoted by $L_f^{||\cdot||}$, is the infimum of all those valid $L$s:

$$L_f^{||\cdot||} := \inf\{L : \forall \mathbf{x}, \mathbf{y} \in \mathcal{X}, |f(\mathbf{x}) - f(\mathbf{y})| \leq L||\mathbf{x} - \mathbf{y}||\}. \tag{1}$$

For deep networks, they play an important role in many applications related to robustness certification which has emerged as an active topic. See recent works [9, 31] based on semidefinite programming (SDP), [8, 40] based on linear programming (LP), [34] based on mixed integer programming (MIP), and [6, 38, 39, 41] based on outer polytope approximation, [7] based on averaged activation operators. We follow a different route and compute upper bounds on the Lipschitz constant of neural networks [35].

Another important application is the Wasserstein Generative Adversarial Network (WGAN) [3]. Wasserstein distance is estimated by using the space of functions encoded by 1-Lipschitz neural networks. This requires a precise estimation of the Lipschitz constants , see recent contributions [2, 11, 25].

Recently there has been a growing interest in polynomial optimization for such problems. In [31], robustness certification is modeled as a quadratically constrained quadratic problem (QCQP) for ReLU networks. Similarly, in [21], an upper bound on the Lipschitz constant of an ELU network is obtained from a polynomial optimization problem (POP). *In contrast to optimization problems with more general functions, powerful (global) positivity certificates are available for POP*. Such certificates *are needed* to approximate global optima as closely as desired [20].

Such positivity certificates have been already applied with success in various areas of science and engineering. The first attempt to compute lower bounds of a QCQP by solving an SDP can be traced back to Shor [32], recently applied to certify robustness of neural networks in [31]. Converging LP based hierarchies are based on Krivine-Stengle's certificates [15, 19, 33]. In [21], sparse versions are used to bound the Lipschitz constant of neural networks. On the other hand, Putinar's certificate [17, 29] is implemented via an SDP-based hierarchy (a.k.a., "Lasserre's hierarchy") and provides converging approximate solutions of a POP. Various applications are described in [16], see also its application to large scale optimal power flow problems in [26, 27] and for roundoff error certification in [24]. The LP hierarchy is cheaper than the SDP hierarchy, but less efficient for combinatorial optimization [22], and cannot converge in finitely many steps for continuous POPs. Finally, weaker positivity certificates can be used, for example DSOS/SDSOS [1] based on second-order cone programming, or hybrid BSOS hierarchy [19].

## 1.1 Related Works

Upper bound on Lipschitz constants of deep networks can be obtained by a product of the layer-wise Lipschitz constants [13]. This is however extremely loose and has many limitations [13]. Note that [36] propose an improvement via a finer product.

Departing from this approach, [21] propose a QCQP formulation to estimate the Lipschitz constant of neural networks. Shor's relaxation allows to obtain a valid upper bound. Alternatively, using the LP hierarchy, [21] obtain tighter upper bounds. By another SDP-based method, [10] provide an upper bound of the Lipschitz constant. However this method is restricted to the $L_2$-norm whereas most robustness certification problems in deep learning are rather concerned with the $L_\infty$-norm.

## 1.2 Preliminaries and Notations

Denote by $F$ the multiple layer neural network, $m$ the number of hidden layers, $p_0, p_1, \ldots, p_m$ the number of nodes in the input layer and each hidden layer. For simplicity, $(p_0, p_1, \ldots, p_m)$ will denote the layer structure of network $F$. Let $\mathbf{x}_0$ be the initial input, and $\mathbf{x}_1, \ldots, \mathbf{x_m}$ be the activation vectors in each hidden layer. Each $\mathbf{x}_i$, $i = 1, \ldots, m$, is obtained by a weight $\mathbf{A}_i$, a bias $\mathbf{b}_i$, and an activation function $\sigma$, i.e., $\mathbf{x}_i = \sigma(\mathbf{A}_i \mathbf{x}_{i-1} + \mathbf{b}_i)$. We only consider coordinatewise application of the *ReLU* activation function, defined as $\mathrm{ReLU}(x) = \max\{0, x\}$ for $x \in \mathbb{R}$. The ReLU function is non-smooth, we define its generalized derivative as the set-valued function $G(x)$ such that $G(x) = 1$ for $x > 0$, $G(x) = 0$ for $x < 0$ and $G(x) = \{0, 1\}$ for $x = 0$.

We assume that the last layer in our neural network is a softmax layer with $K$ entries, that is, the network is a classifier for $K$ labels. For each label $k \in \{1, \ldots, K\}$, the score of label $k$ is obtained by an affine product with the last activation vector, i.e., $\mathbf{c}_k^T \mathbf{x}_m$ for some $\mathbf{c}_k \in \mathbb{R}^{p_m}$. The final output is the label with the highest score, i.e., $y = \arg\max_k \mathbf{c}_k^T \mathbf{x}_m$. The product $\mathbf{xy}$ of two vectors $\mathbf{x}$ and $\mathbf{y}$ is considered as coordinate-wise product.

## 1.3 Contribution

• We first express both graphs of $\mathrm{ReLU}$ and its generalized derivative via *basic closed semialgebraic* sets, i.e., sets defined with finite conjunctions of polynomial (in)equalities. Indeed, if $y = \mathrm{ReLU}(x)$ and $v \in G(x)$, then equivalently: $y(y - x) = 0, y \geq x, y \geq 0$ and $v(v - 1) = 0, (v - 1/2)x \geq 0$. Note that the explicit semialgebraic expression of ReLU has already been used in related works, see for example [31]. The semialgebraic expression for $G$ is our contribution. Being *exact*, this semi-algebraic reformulation is a noticeable improvement compared to the model proposed in [21] where the generalized derivative of $\mathrm{ReLU}$ is simply replaced with a decision variable lying between 0 and 1 (and so is only approximated).

Table 1: Global Lipschitz constant and solver running time of networks of size $(80, 80)$ and $(40, 40, 10)$ obtained by **HR-1**, **HR-2**, **LipOpt-3** and **LipOpt-4** on various sparsities $s$. The abbreviation "**HR-2**" (resp. "**HR-1**") stands for the second-order (resp. first-order) SDP-based method we propose (see Appendix E), and "**LipOpt-3/4**" stands for the LP-based method from [21] (which uses Krivine-Stengle's positivity certificate of degree 3 or 4). **LBS** is the lower bound computed by random sampling. For networks of more than 2 hidden layers, we use the technique introduced in Appendix E in order to deal with the cubic terms in the objective. *OfM* means out of memory while building the model. The results are for a single random network, complete results are shown in Appendix F and G.

| | | (80, 80) | | | | | (40, 40, 10) | | | |
|---|---|---|---|---|---|---|---|---|---|---|
| | | $s=20$ | $s=40$ | $s=60$ | $s=80$ | | $s=20$ | $s=40$ | $s=60$ | $s=80$ |
| **HR-2** | OBJ. | 1.45 | 2.05 | 2.41 | 2.68 | **HR-1** | 0.50 | 1.16 | 1.82 | 2.05 |
| | TIME | 3.14 | 7.78 | 8.61 | 9.82 | | 271.34 | 165.68 | 174.86 | 174.02 |
| **LIPOPT-3** | OBJ. | 1.55 | 2.86 | 3.85 | 4.68 | **LIPOPT-3** | 0.56 | 1.68 | 3.01 | 3.57 |
| | TIME | 2.44 | 10.36 | 20.99 | 71.49 | | 3.84 | 4.83 | 7.91 | 6.33 |
| **LIPOPT-4** | OBJ. | 1.43 | OFM | OFM | OFM | **LIPOPT-4** | 0.29 | 0.85 | OFM | OFM |
| | TIME | 127.99 | OFM | OFM | OFM | | 321.89 | 28034.27 | OFM | OFM |
| **LBS** | OBJ. | 1.05 | 1.56 | 1.65 | 1.86 | **LBS** | 0.20 | 0.48 | 0.61 | 0.62 |

• Second, we provide a heuristic approach based on an SDP-hierarchy for nearly sparse polynomial optimization problems. In such problems one assumes that only a few affine constraints destroy a sparsity pattern satisfied by the other constraints. This new approach is mainly based on the sparse version of the Lasserre's hierarchy, popularized in [18, 37]. It provides upper bounds yielding a trade-off between third and fourth degree LP relaxations in [21], and sometimes a strict improvement.

## 1.4 Main Results

In recent work [21] a certain sparsity structure arising from a neural network is exploited. Consider a neural network $F$ with one single hidden layer, and 4 nodes in each layer. The network $F$ is said to have a sparsity of 4 if its weight matrix $\mathbf{A}$ is symmetric with diagonal blocks of size at most $2 \times 2$:

$$\begin{pmatrix} * & * & 0 & 0 \\ * & * & * & 0 \\ 0 & * & * & * \\ 0 & 0 & * & * \end{pmatrix} \tag{2}$$

Larger sparsity values refer to symmetric matrices with band structure of a given size. This sparsity structure (2) of the networks greatly influences the number of variables involved in the LP program to solve in [21]. This is in deep contrast with our method which does not require the weight matrix to be as in (2). Hence when the network is fully-connected, our method is more efficient and provides tighter upper bounds.

Table 1 gives a brief comparison outlook of the results obtained by our method and the method in [21]. For $(80, 80)$ networks, apart from $s = 20$, which is not significant, **HR-2** obtains much better bounds and is also much more efficient than **LipOpt-3**. **LipOpt-4** provides tighter bounds than **HR-2** but suffers more computational time, and run out of memory when the sparsity increases. For $(40, 40, 10)$ networks, **HR-1** is a trade-off between **LipOpt-3** and **LipOpt-4**, it provides tighter (resp. looser) bounds than **LipOpt-3** (resp. **LipOpt-4**), but takes more (resp. less) computational time.

## 2 Problem Setting

In this section, we recall basic facts about optimization and build the polynomial optimization model for estimating Lipschitz constant of neural networks.

### 2.1 Polynomial Optimization

In a *polynomial optimization problem (POP)*, one computes the *global* minimum (or maximum) of a multivariate polynomial function on a *basic closed semialgebraic* set. If the semialgebraic set is the whole space, the problem is *unconstrained*, and *constrained* otherwise. Given a positive integer $n \in \mathbb{N}$, let $\mathbf{x} = (x_1, \dots, x_n)^T$ be a vector of decision variables, and denote by $[n]$ the set $\{1, \dots, n\}$.

A POP has the canonical form:

$$\inf_{\mathbf{x}\in\mathbb{R}^n} \{f(\mathbf{x}) : f_i(\mathbf{x}) \geq 0, i \in [p]; g_j(\mathbf{x}) = 0, j \in [q]\}, \tag{POP}$$

where $f, f_i, g_j$ are all polynomials in $n$ variables. With $I \subset \{1, \ldots, n\}$, let $\mathbf{x}_I := (x_i)_{i \in I}$ and let $\mathbb{R}[\mathbf{x}]$ be the space of real polynomials in the variables $\mathbf{x}$ while $\mathbb{R}[\mathbf{x}_I]$ is the space of real polynomials in the variables $\mathbf{x}_I$.

In particular, if the objective $f$ and constraints $f_i, g_j$ in (POP) are all of degree at most 2, we say that the problem is a *quadratically constrainted quadratic problem (QCQP)*. The Shor's relaxation of a QCQP is a semidefinite program which can be solved efficiently numerically. If all polynomials involved in (POP) are affine then the problem is a *linear program (LP)*.

## 2.2 Lipschitz Constant Estimation Problem (LCEP)

Suppose we train a neural network $F$ for $K$-classifications and denote by $\mathbf{A}_i, \mathbf{b}_i, \mathbf{c}_k$ its parameters already defined in section 1.2. Thus for an input $\mathbf{x}_0 \in \mathbb{R}^{p_0}$, the targeted score of label $k$ can be expressed as $F_k(\mathbf{x}_0) = \mathbf{c}_k^T \mathbf{x}_m$, where $\mathbf{x}_i = \mathrm{ReLU}(\mathbf{A}_i \mathbf{x}_{i-1} + \mathbf{b}_i)$, for $i \in [m]$. Let $\mathbf{z}_i = \mathbf{A}_i \mathbf{x}_{i-1} + \mathbf{b}_i$ for $i \in [m]$. By applying the chain rule on the non-smooth function $F_k$, we obtain a set valued map for $F_k$ at point any $\mathbf{x}_0$ as $G_{F_k}(\mathbf{x}_0) = (\prod_{i=1}^m \mathbf{A}_i^T \mathrm{diag}(G(\mathbf{z}_i)))\mathbf{c}_k$.

We fix a targeted label (label 1 for example) and omit the symbol $k$ for simplicity. We define $L_F^{||\cdot||}$ of $F$ with respect to norm $||\cdot||$, is the supremum of the gradient's dual norm, i.e.:

$$L_F^{||\cdot||} = \sup_{\mathbf{x}_0 \in \Omega, \mathbf{v} \in G_{F_k}(\mathbf{x}_0)} ||\mathbf{v}||_* = \sup_{\mathbf{x}_0 \in \Omega} \left|\left|\left(\prod_{i=1}^m \mathbf{A}_i^T \mathrm{diag}(G(\mathbf{z}_i))\right)\mathbf{c}\right|\right|_*, \tag{3}$$

where $\Omega$ is the convex input space, and $||\cdot||_*$ is the dual norm of $||\cdot||$, which is defined by $||\mathbf{x}||_* := \sup_{||\mathbf{t}|| \leq 1} |\langle \mathbf{t}, \mathbf{x}\rangle|$ for all $\mathbf{x} \in \mathbb{R}^n$. In general, the chain rule cannot be applied to composition of non-smooth functions [5, 14]. Hence the formulation of $G_{F_k}$ and (3) may lead to incorrect gradients and bounds on the Lipschitz constant of the networks. The following ensures that this is not the case and that the approach is sound, its proof is postponed to Appendix A.

**Lemma 1** *If $\Omega$ is convex, then $L_F^{||\cdot||}$ is a Lipschitz constant for $F_k$ on $\Omega$.*

When $\Omega = \mathbb{R}^n$, $L_F^{||\cdot||}$ is the *global* Lipschitz constant of $F$ with respect to norm $||\cdot||$. In many cases we are also interested in the *local* Lipschitz constant of a neural network constrained in a small neighborhood of a fixed input $\bar{\mathbf{x}}_0$. In this situation the input space $\Omega$ is often the ball around $\bar{\mathbf{x}}_0 \in \mathbb{R}$ with radius $\varepsilon$: $\Omega = \{\mathbf{x} : ||\mathbf{x} - \bar{\mathbf{x}}_0|| \leq \varepsilon\}$. In particular, with the $L_\infty$-norm (and using $l \leq x \leq u \Leftrightarrow (x-l)(x-u) \leq 0$), the input space $\Omega$ is the basic semialgebraic set:

$$\Omega = \{\mathbf{x} : (\mathbf{x} - \bar{\mathbf{x}}_0 + \varepsilon)(\mathbf{x} - \bar{\mathbf{x}}_0 - \varepsilon) \leq 0\}. \tag{4}$$

Combining Lemma 1 and (3), the *Lipschitz constant estimation problem (LCEP)* for neural networks with respect to the norm $||\cdot||$, is the following POP:

$$\max_{\mathbf{x}_i, \mathbf{u}_i, \mathbf{t}} \{\mathbf{t}^T \left(\prod_{i=1}^m \mathbf{A}_i^T \mathrm{diag}(\mathbf{u}_i)\right)\mathbf{c} : \mathbf{u}_i(\mathbf{u}_i - 1) = 0, (\mathbf{u}_i - 1/2)(\mathbf{A}_i \mathbf{x}_{i-1} + \mathbf{b}_i) \geq 0, i \in [m];$$

$$\mathbf{x}_{i-1}(\mathbf{x}_{i-1} - \mathbf{A}_{i-1}\mathbf{x}_{i-2} - \mathbf{b}_{i-1}) = 0, \mathbf{x}_{i-1} \geq 0, \mathbf{x}_{i-1} \geq \mathbf{A}_{i-1}\mathbf{x}_{i-2} + \mathbf{b}_{i-1}, 2 \leq i \leq m;$$

$$\mathbf{t}^2 \leq 1, (\mathbf{x}_0 - \bar{\mathbf{x}}_0 + \varepsilon)(\mathbf{x}_0 - \bar{\mathbf{x}}_0 - \varepsilon) \leq 0.\} \tag{LCEP}$$

In [21] the authors only use the constraint $0 \leq \mathbf{u}_i \leq 1$ on the variables $\mathbf{u}_i$, only capturing the Lipschitz character of the considered activation function. We could use the same constraints, this would allow to use activations which do not have semi-algebraic representations such as the Exponential Linear Unit (ELU). However, such a relaxation, despite very general, is a lot coarser than the one we propose. Indeed, (LCEP) treats an *exact formulation* of the generalized derivative of the ReLU function by exploiting its semialgebraic character.

## 3 Lasserre's Hierarchy

In this section we briefly introduce the Lasserre's hierarchy [17] which has already many successful applications in and outside optimization [20].

## 3.1 Convergent SDP Relaxations without Exploiting Sparsity [17]

In the Lasserre's hierarchy for optimization one approximates the *global* optimum of the POP

$$f^* = \inf_{\mathbf{x} \in \mathbb{R}^n} \{ f(\mathbf{x}) : g_i(\mathbf{x}) \geq 0, i \in [p] \}, \tag{Opt}$$

(where $f, g_i$ are all polynomials in $\mathbb{R}[\mathbf{x}]$), by solving a hierarchy of SDPs [1] of increasing size. Equality constraints can also be taken into account easily. Each SDP is a semidefinite relaxation of (Opt) in the form:

$$\rho_d = \inf_{\mathbf{y}} \{ L_{\mathbf{y}}(f) : L_{\mathbf{y}}(1) = 1, \mathbf{M}_d(\mathbf{y}) \succeq 0, \mathbf{M}_{d-\omega_i}(g_i \mathbf{y}) \succeq 0, i \in [p] \}, \tag{MomOpt-$d$}$$

where $\omega_i = \lceil \deg(g_j)/2 \rceil$, $\mathbf{y} = (y_\alpha)_{\alpha \in \mathbb{N}^n_{2d}}$, $L_{\mathbf{y}} : \mathbb{R}[\mathbf{x}] \to \mathbb{R}$ is the so-called *Riesz linear functional*: $f = \sum_\alpha f_\alpha \mathbf{x}^\alpha \mapsto L_{\mathbf{y}}(f) := \sum_\alpha f_\alpha y_\alpha$ with $f \in \mathbb{R}[\mathbf{x}]$, and $\mathbf{M}_d(\mathbf{y})$, $\mathbf{M}_{d-\omega_i}(g_i \mathbf{y})$ are *moment matrix* and *localizing matrix* respectively; see [20] for precise definitions and more details. The semidefinite program (MomOpt-$d$) is the $d$-th order *moment relaxation* of problem (Opt). As a result, when $\mathbf{K} := \{ \mathbf{x} : g_i(\mathbf{x}) \geq 0, i \in [p] \}$ is compact, one obtains a monotone sequence of lower bounds $(\rho_d)_{d \in \mathbb{N}}$ with the property $\rho_d \uparrow f^*$ as $d \to \infty$ under a certain technical Archimedean condition; the latter is easily satisfied by including a quadratic redundant constraint $M - \|\mathbf{x}\|^2 \geq 0$ in the definition of $\mathbf{K}$ (redundant as $\mathbf{K}$ is compact and $M$ is large enough). At last but not least and interestingly, generically the latter convergence is *finite* [28]. Ideally, one expects an optimal solution $\mathbf{y}^*$ of (MomOpt-$d$) to be the vector of moments up to order $2d$ of the Dirac measure $\delta_{\mathbf{x}^*}$ at a global minimizer $\mathbf{x}^*$ of (Opt). See Appendix B.1 for an example to illustrate the core idea of Lasserre's hierarchy.

## 3.2 Convergent SDP Relaxations Exploiting Sparsity [18, 37]

The hierarchy (MomOpt-$d$) is often referred to as *dense* Lasserre's hierarchy since we do not exploit any possible sparsity pattern of the POP. Therefore, if one solves (MomOpt-$d$) with interior point methods (as all current SDP solvers do), then the dense hierarchy is limited to POPs of modest size. Indeed the $d$-th order dense moment relaxation (MomOpt-$d$) involves $\binom{n+2d}{2d}$ variables and a moment matrix $M_d(\mathbf{y})$ of size $\binom{n+d}{d} = O(n^d)$ at fixed $d$. Fortunately, large-scale POPs often exhibit some structured sparsity patterns which can be exploited to yield a *sparse version* of (MomOpt-$d$), as initially demonstrated in [37]. As a result, wider applications of Lasserre's hierarchy have been possible.

Assume that the set of variables in (Opt) can be divided into several subsets indexed by $I_k$, for $k \in [l]$, i.e., $[n] = \cup_{k=1}^l I_k$, and suppose that the following assumptions hold:

**A1**: The function $f$ is a sum of polynomials, each involving variables of only one subset, i.e., $f(\mathbf{x}) = \sum_{k=1}^l f_k(\mathbf{x}_{I_k})$;
**A2**: Each constraint also involves variables of only one subset, i.e., $g_i \in \mathbb{R}[\mathbf{x}_{I_{k(i)}}]$ for some $k(i) \in [l]$;
**A3**: The subsets $I_k$ satisfy the *Running Intersection Property (RIP)*: for every $k \in [l-1]$, $I_{k+1} \cap \bigcup_{j=1}^k I_j \subseteq I_s$, for some $s \leq k$.
**A4**: Add redundant constraints $M_k - \|\mathbf{x}_{I_k}\|^2 \geq 0$ where $M_k$ are constants determined beforehand.

A POP with such a sparsity pattern is of the form:

$$\inf_{\mathbf{x} \in \mathbb{R}^n} \{ f(\mathbf{x}) : g_i(\mathbf{x}_{I_{k(i)}}) \geq 0, \ i \in [p] \}, \tag{SpOpt}$$

and its associated *sparse Lasserre's hierarchy* reads:

$$\theta_d = \inf_{\mathbf{y}} \{ L_{\mathbf{y}}(f) : L_{\mathbf{y}}(1) = 1, \mathbf{M}_d(\mathbf{y}, I_k) \succeq 0, k \in [l]; \mathbf{M}_{d-\omega_i}(g_i \, \mathbf{y}, I_{k(i)}) \succeq 0, \ i \in [p] \},$$

$$\tag{MomSpOpt-$d$}$$

where $d, \omega_i, \mathbf{y}, L_{\mathbf{y}}$ are defined as in (MomOpt-$d$) but with a crucial difference. The matrix $\mathbf{M}_d(\mathbf{y}, I_k)$ (resp. $\mathbf{M}_{d-\omega_i}(g_i \, \mathbf{y}, I_k)$) is a submatrix of the moment matrix $\mathbf{M}_d(\mathbf{y})$ (resp. localizing matrix

$\mathbf{M}_{d-\omega_i}(g_i\mathbf{y}))$ with respect to the subset $I_k$, and hence of much smaller size $\binom{\tau_k+d}{\tau_k}$ if $|I_k| =: \tau_k \ll n$. See Appendix B.2 for an example to illustrate the core idea of sparse Lasserre's hierarchy.

If the maximum size $\tau$ of the subsets is such that $\tau \ll n$, then solving (MomSpOpt-$d$) rather than (MomOpt-$d$) results in drastic computational savings. In fact, even with not so large $n$, (MomOpt-$d$) the second relaxation with $d = 2$ is out of reach for currently available SDP solvers. Finally, $\theta_d \le f^*$ for all $d$ and moreover, if the subsets $I_k$ satisfy RIP, then we still obtain the convergence $\theta_d \uparrow f^*$ as $d \to \infty$, like for the dense relaxation (MomOpt-$d$).

There is a primal-dual relation between the moment problem and the sum-of-square (SOS) problem, as shown in Appendix C. The specific MATLAB toolboxes Gloptipoly [12] and YALMIP [23] can solve the hierarchy (MomOpt-$d$) and its sparse variant (MomSpOpt-$d$), and also their dual SOS problem.

## 4  Heuristic Approaches

For illustration purpose, consider 1-hidden layer networks. Then in (LCEP) we can define natural subsets $I_i = \{u_1^{(i)}, \mathbf{x}_0\}$, $i \in [p_1]$ (w.r.t. constraints $\mathbf{u}_1(\mathbf{u}_1 - 1) = 0$, $(\mathbf{u}_1 - 1/2)(\mathbf{A}_1\mathbf{x}_0 + \mathbf{b}_1) \ge 0$, and $(\mathbf{x}_0 - \bar{\mathbf{x}}_0 + \varepsilon)(\mathbf{x}_0 - \bar{\mathbf{x}}_0 - \varepsilon) \le 0$); and $J_j = \{t^{(j)}\}$, $j \in [p_0]$ (w.r.t. constraints $\mathbf{t}^2 \le 1$). Clearly, $I_i, J_j$ satisfy the RIP condition and are subsets with smallest possible size. Recall that $\mathbf{x}_0 \in \mathbb{R}^{p_0}$. Hence $|I_i| = 1 + p_0$ and the maximum size of the PSD matrices is $\binom{1+p_0+d}{d}$. Therefore, as in real deep neural networks $p_0$ can be as large as 1000, the second-order sparse Lasserre's hierarchy (MomSpOpt-$d$) cannot be implemented in practice.

In fact (LCEP) can be considered as a "nearly sparse" POP, i.e., a sparse POP with some additional "bad" constraints that violate the sparsity assumptions. More precisely, suppose that $f, g_i$ and subsets $I_k$ satisfy assumptions **A1**, **A2**, **A3** and **A4**. Let $g$ be a polynomial that violates **A2**. Then we call the POP

$$\inf_{\mathbf{x}\in\mathbb{R}^n} \{f(\mathbf{x}) : g(\mathbf{x}) \ge 0, g_i(\mathbf{x}) \ge 0, \ i \in [p]\}, \qquad \text{(NlySpOpt)}$$

a *nearly sparse* POP because only one constraint, namely $g \ge 0$, does not satisfy the sparsity pattern **A2**. This single "bad" constraint $g \ge 0$ precludes us from applying the sparse Lasserre hierarchy (MomSpOpt-$d$).

In this situation, we propose a heuristic method which can be applied to problems with arbitrary many constraints that possibly destroy the sparsity. The key idea of our algorithm is: **(i)** Keep the "nice" sparsity pattern defined without the bad constraints; **(ii)** Associate only low-order localizing matrix constraints to the "bad" constraints. In brief, the $d$-th order *heuristic hierarchy* (*HR-$d$*) reads:

$$\inf_{\mathbf{y}}\{L_{\mathbf{y}}(f) : \mathbf{M}_1(\mathbf{y}) \succeq 0, \mathbf{M}_d(\mathbf{y}, I_k) \succeq 0, k \in [l]; \mathbf{M}_{d-\omega_i}(g_i\,\mathbf{y}, I_{k(i)}) \succeq 0, \ i \in [p];$$

$$L_{\mathbf{y}}(g) \ge 0, L_{\mathbf{y}}(1) = 1\}, \qquad \text{(MomNlySpOpt-$d$)}$$

where $\mathbf{y}, L_{\mathbf{y}}, \mathbf{M}_d(\mathbf{y}, I_k), \mathbf{M}_{d-\omega_i}(g_i\mathbf{y}, I_{k(i)})$ have been defined in section 3.2. For more illustration of this heuristic relaxation and how it is applied to estimate the Lipschitz constant of neural networks, see Appendix D.

For simplicity, assume that the neural networks have only one single hidden layer, i.e., $m = 1$. Denote by $A, b$ the weight and bias respectively. As in (4), we use the fact that $l \le x \le u$ is equivalent to $(x - l)(x - u) \le 0$. Then the local Lipschitz constant estimation problem with respect to $L_\infty$-norm can be written as:

$$\max_{\mathbf{x},\mathbf{u},\mathbf{z},\mathbf{t}}\{\mathbf{t}^T \mathbf{A}^T \mathrm{diag}(\mathbf{u})\mathbf{c} : (\mathbf{z} - \mathbf{A}\mathbf{x} - \mathbf{b})^2 = 0, \mathbf{t}^2 \le 1, (\mathbf{x} - \bar{\mathbf{x}}_0 + \varepsilon)(\mathbf{x} - \bar{\mathbf{x}}_0 - \varepsilon) \le 0,$$

$$\mathbf{u}(\mathbf{u} - 1) = 0, (\mathbf{u} - 1/2)\mathbf{z} \ge 0\}. \qquad \text{(LCEP-MLP}_1)$$

Define the subsets of (LCEP-MLP$_1$) to be $I^i = \{x^i, t^i\}$, $J^j = \{u^j, z^j\}$ for $i \in [p_0]$, $j \in [p_1]$, where $p_0, p_1$ are the number of nodes in the input layer and hidden layer respectively. Then the second-order

$(d = 2)$ heuristic relaxation of (LCEP-MLP$_1$) is the following SDP:

$$\inf_{\mathbf{y}}\{L_{\mathbf{y}}(\mathbf{t}^T \mathbf{A}^T \mathrm{diag}(\mathbf{u})\mathbf{c}) : L_{\mathbf{y}}(1) = 1, \mathbf{M}_1(\mathbf{y}) \succeq 0, L_{\mathbf{y}}(\mathbf{z} - \mathbf{Ax} - \mathbf{b}) = 0, L_{\mathbf{y}}((\mathbf{z} - \mathbf{Ax} - \mathbf{b})^2) = 0;$$

$$\mathbf{M}_2(\mathbf{y}, I^i) \succeq 0, \mathbf{M}_1(-(x^{(i)} - \bar{x}_0^{(i)} + \varepsilon)(x^{(i)} - \bar{x}_0^{(i)} - \varepsilon)\mathbf{y}, I^i) \succeq 0, \mathbf{M}_1((1 - t_i^2)\mathbf{y}, I^i) \succeq 0, i \in [p_0];$$

$$\mathbf{M}_2(\mathbf{y}, J^j) \succeq 0, \mathbf{M}_1(u_j(u_j - 1)\mathbf{y}, J^j) = 0, \mathbf{M}_1((u_j - 1/2)z_j\mathbf{y}, J^j) \succeq 0, j \in [p_1].\}.$$
(MomLCEP-2)

The $d$-th order heuristic relaxation (MomNlySpOpt-$d$) also applies to multiple layer neural networks. However, if the neural network has $m$ hidden layers, then the criterion in (LCEP) is of degree $m + 1$. If $m \geq 2$, then the first-order moment matrix $\mathbf{M}_1(\mathbf{y})$ is no longer sufficient, as moments of degree $> 2$ are *not* encoded in $\mathbf{M}_1(\mathbf{y})$ and some may not be encoded in the moment matrices $\mathbf{M}_2(\mathbf{y}, I^i)$, if they include variables of different subsets. See Appendix E for more information to deal with higher-degree polynomial objective.

## 5 Experiments

In this section, we provide results for the *global* and *local* Lipschitz constants of *random* networks of fixed size $(80, 80)$ and with various sparsities. We also compute bounds of a *real* trained 1-hidden layer network. The complete results for global/local Lipschitz constants of both 1-hidden layer and 2-hidden layer networks can be found in Appendix F and G. For all experiments we focus on the $L_\infty$-norm, the most interesting case for robustness certification. Moreover, we use the Lipschitz constants computed by various methods to certify robustness of a trained network, and compare the ratio of certified inputs with different methods. Let us first provide an overview of the methods with which we compare our results.

**SHOR**: Shor's relaxation applied to (LCEP). Note that this is different from Shor's relaxation decribed in [21] since we apply it to a different QCQP.
**HR-2**: second-order heuristic relaxation applied to (LCEP).
**LipOpt-3**: LP-based method by [21] with degree 3.
**LBS**: lower bound obtained by sampling 50000 random points and evaluating the dual norm of the gradient.

The reason why we list **LBS** here is because **LBS** is a valid lower bound on the Lipschitz constant. Therefore all methods should provide a result not lower than **LBS**, a basic necessary condition of consistency.

As discussed in section 2.2, if we want to estimate the global Lipschitz constant, we need the input space $\Omega$ to be the whole space. In consideration of numerical issues, we set $\Omega$ to be the ball of radius 10 around the origin. For the local Lipschitz constant, we set by default the radius of the input ball as $\varepsilon = 0.1$. In both cases, we compute the Lipschitz constant with respect to the first label. All experiments are run on a personal laptop with a 4-core i5-6300HQ 2.3GHz CPU and 8GB of RAM. We use the (Python) code provided by [21][2] to execute the experiments for **LipOpt** with Gurobi solver. For **HR-2** and **SHOR**, we use the YALMIP toolbox (MATLAB) [23] with MOSEK as a backend to calculate the Lipschitz constants for *random* networks. For *trained* network, we implement our algorithm on Julia [4] with MOSEK optimizer to accelerate the computation.

*Remark*: The crossover option [3] in Gurobi solver is activated by default, and it is used to transform the interior solution produced by barrier into a basic solution. We deactivate this option in our experiments since this computation is unnecessary and takes a lot of time. Throughout this paper, running time is referred to the time taken by the LP/SDP solver (Gurobi/Mosek) and *OfM* means running out of memory during building up the LP/SDP model.

### 5.1 Lipschitz Constant Estimation

**Random Networks.** We first compare the upper bounds for $(80, 80)$ networks, whose weights and biases are randomly generated. We use the codes provided by [21] to generate networks with various sparsities. For each fixed sparsity, we generate 10 different random networks, and apply all the

Table 2: Comparison of upper bounds of global Lipschitz constant and solver running time on trained network SDP-NN obtained by **HR-2**, **SHOR**, **LipOpt-3** and **LBS**. The network is a fully connected neural network with one hidden layer, with 784 nodes in the input layer and 500 nodes in the hidden layer. The network is for 10-classification, we calculate the upper bound with respect to label 2.

| | GLOBAL | | | | LOCAL | | | |
| | **HR-2** | **SHOR** | **LIPOPT-3** | **LBS** | **HR-2** | **SHOR** | **LIPOPT-3** | **LBS** |
|---|---|---|---|---|---|---|---|---|
| BOUND | 14.56 | 17.85 | OFM | 9.69 | 12.70 | 16.07 | OFM | 8.20 |
| TIME | 12246 | 2869 | OFM | - | 20596 | 4217 | OFM | - |

methods to them repeatedly. Then we compute the average upper bound and average running time of those 10 experiments.

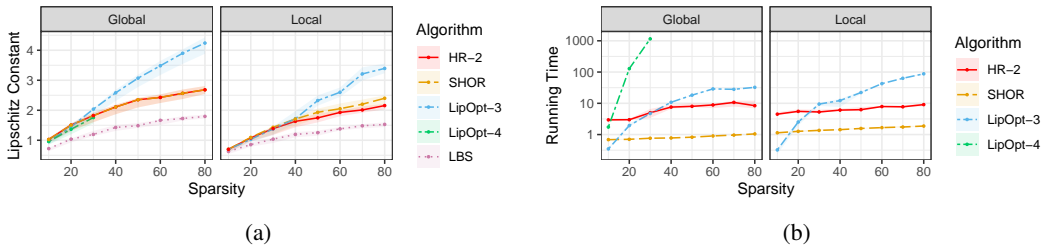

(a)                                          (b)

Figure 1: Lipschitz constant upper bounds and solver running time with respect to $L_\infty$ norm obtained by **HR-2**, **SHOR**, **LipOpt-3**, **LipOpt-4** and **LBS**. We generate random networks of size $(80, 80)$ with sparsity 10, 20, 30, 40, 50, 60, 70, 80. In the meantime, we display median and quartiles over 10 random networks draws.

Figure 1 displays a comparison of average upper bounds of global and local Lipschitz constants. For global bounds, we can see from Figure 1a that when the sparsity of the network is small (10, 20, etc.), the LP-based method **LipOpt-3** is slightly better than the SDP-based method **HR-2**. As the sparsity increases, **HR-2** provides tighter bounds. Figure 1b shows that **LipOpt-3** is more efficient than **HR-2** only for sparsity 10. When the networks are dense or nearly dense, our method not only takes much less time, but also gives tighter upper bounds. For global Lipschitz constant estimation, **SHOR** and **HR-2** give nearly the same upper bounds. However, in the local case, **HR-2** provides strictly tighter bounds than **SHOR**. In both global and local cases, **SHOR** has smaller computational time than **HR-2**. In Appendix F and G, we present more results of global and local Lipschitz constant bounds for networks of various sizes and sparsities.

**Trained Network** We use the MNIST classifier (*SDP-NN*) described in [30][4]. The network is of size $(784, 500)$. In Table 2, we see that the **LipOpt-3** algorithm runs out of memory when applied to the real network SDP-NN to compute the global Lipschitz bound. In contrast, **SHOR** and **HR-2** still work and moreover, **HR-2** provides tighter upper bounds than **SHOR** in both global and local cases. As a trade-off, the running time of **HR-2** is around 5 times longer than that of **SHOR**.

### 5.2 Robustness Certification

**Binary Classifier** We generate 20000 random points in dimension 80, where half of them are concentrated around the sphere with radius 1 (labeled as 1) and the rest are concentrated around the sphere with radius 2 (labeled as 2). We train a (80, 80) network for this binary classification task, and use *sigmoid* layer as the output layer. Therefore, the input is labeled as 1 (resp. 2) if the output is negative (resp. positive). For an input $\mathbf{x}_0$ and a perturbation $\epsilon$, we compute an upper bound of the Lipschitz constant with respect to $\mathbf{x}_0$, $\epsilon$ and $L_\infty$-norm, denoted by $L^{\mathbf{x}_0, \epsilon}$. Then, if the output $y_0$ is negative (resp. positive) and $y_0 + \epsilon L^{\mathbf{x}_0, \epsilon} < 0$ (resp. $y_0 - \epsilon L^{\mathbf{x}_0, \epsilon} > 0$), the input $\mathbf{x}_0$ is $\epsilon$-robust. With this criteria, if we have $n$ inputs to certify, then we need to execute $n$ experiments.

Table 3: Comparison of ratios of certified examples for $(80, 80)$ network by **HR-2** and **LipOpt-3**.

| $\epsilon$ | 0.01 | 0.02 | 0.03 | 0.04 | 0.05 | 0.06 | 0.07 | 0.08 | 0.09 | 0.1 |
|---|---|---|---|---|---|---|---|---|---|---|
| **HR-2** | 87.51% | 75.02% | 62.46% | 49.89% | 37.22% | 24.36% | 8.15% | 2.75% | 0.76% | 0.12% |
| **LipOpt-3** | 69.03% | 37.84% | 4.78% | 0.15% | 0% | 0% | 0% | 0% | 0% | 0% |

Table 4: Ratios of certified test examples for SDP-NN network by **HR-2**.

| $\epsilon$ | 0.01 | 0.02 | 0.03 | 0.04 | 0.05 | 0.06 | 0.07 | 0.08 | 0.09 | 0.1 |
|---|---|---|---|---|---|---|---|---|---|---|
| Ratios | 98.80% | 97.24% | 95.16% | 92.84% | 90.18% | 87.10% | 83.01% | 78.34% | 73.54% | 67.63% |

However, for large networks such as MNIST networks, this is impractical, even if we certify the network directly without using Lipschitz constants (see [30]). Therefore, we compute the Lipschitz constant with respect to $\mathbf{x}_0 = \mathbf{0}$ and $\epsilon = 3$, denoted by $L$, and generate $10^6$ random points in the box $\mathbf{B} = \{\mathbf{x} : ||\mathbf{x}||_\infty \leq 2.9\}$. For any $\epsilon \leq 0.1$ and $\mathbf{x}_0 \in \mathbf{B}$, we have $L^{\mathbf{x}_0,\epsilon} \leq L$, then the point $\mathbf{x}_0$ is $\epsilon$-robust if $y_0(y_0 - \text{sign}(y_0)\epsilon L) > 0$. Instead of running $10^6$ times the algorithm, we are able to certify robustness of large number of inputs by only doing one experiment. Table 3 shows the ratio of certified points in the box $\mathbf{B}$ by **HR-2** and **LipOpt-3**. We see that **HR-2** can always certify more examples than **LipOpt-3**.

**Multi-Classifier** As described in Section 5.1, the SDP-NN network is a well-trained $(784, 500)$ network to classify the digit images from 0 to 9. Denote the parameters of this network by $\mathbf{A} \in \mathbb{R}^{500 \times 784}, \mathbf{b}_1 \in \mathbb{R}^{500}, \mathbf{C} \in \mathbb{R}^{10 \times 500}, \mathbf{b}_2 \in \mathbb{R}^{10}$. The score of an input $\mathbf{x}$ is denoted by $\mathbf{y}^{\mathbf{x}}$, i.e., $\mathbf{y}^{\mathbf{x}} = \mathbf{C} \cdot \text{ReLU}(\mathbf{A}\mathbf{x}_0 + \mathbf{b}_1) + \mathbf{b}_2$. The label of $\mathbf{x}$, denoted by $r^{\mathbf{x}}$, is the index with the largest score, i.e., $r^{\mathbf{x}} = \arg\max \mathbf{y}^{\mathbf{x}}$. Suppose an input $\mathbf{x}_0$ has label $r$. For $\epsilon$ and $\mathbf{x}$ such that $||\mathbf{x} - \mathbf{x}_0||_\infty \leq \epsilon$, if for all $i \neq r, y_i^{\mathbf{x}} - y_r^{\mathbf{x}} < 0$, then $\mathbf{x}_0$ is $\epsilon$-robust. Alternatively, denote by $L_{i,r}^{\mathbf{x}_0,\epsilon}$ the local Lipschitz constant of function $f_{i,r}(\mathbf{x}) = y_i^{\mathbf{x}} - y_r^{\mathbf{x}}$ with respect to $L_\infty$-norm in the ball $\{\mathbf{x} : ||\mathbf{x} - \mathbf{x}_0||_\infty \leq \epsilon\}$. Then the point $\mathbf{x}_0$ is $\epsilon$-robust if for all $i \neq r, f_{i,r}(\mathbf{x}_0) + \epsilon L_{i,r}^{\mathbf{x}_0,\epsilon} < 0$. Since the $28 \times 28$ MNIST images are flattened and normalized into vectors taking value in $[0, 1]$, we compute the local Lipschitz constant (by **HR-2**) with respect to $\mathbf{x}_0 = \mathbf{0}$ and $\epsilon = 2$, the complete value is referred to matrix $\mathbf{L}$ in Appendix H. We take different values of $\epsilon$ from 0.01 to 0.1, and compute the ratio of certified examples among the 10000 MNIST test data by the Lipschitz constants we obtain, as shown in Table 4. Note that for $\epsilon = 0.1$, we improve a little bit by 67% compared to **Grad-cert** (65%) described in [30], as we use an exact formulation of the derivative of ReLU function.

## 6 Conclusion and Future Work

**Optimization Aspect:** In this work, we propose a new heuristic moment relaxation based on the dense and sparse Lasserre's hierarchy. In terms of performance, our method provides bounds no worse than Shor's relaxation and better bounds in many cases. In terms of computational efficiency, our algorithm also applies to nearly sparse polynomial optimization problems without running into computational issue.

**Machine Learning Aspect:** The ReLU function and its generalized derivative $G(x)$ are semialgebraic. This semialgebraic character is easy to handle *exactly* in polynomial optimization (via some lifting) so that one is able to apply moment relaxation techniques to the resulting POP. Moreover, our heuristic moment relaxation provides tighter bounds than Shor's relaxation and the state-of-the-art LP-based algorithm in [21].

**Future research:** The heuristic relaxation is designed for QCQP (e.g. problem (LCEP) for 1-hidden layer networks). As the number of hidden layer increases, the degree of the objective function also increases and the approach must be combined with *lifting* or sub-moment techniques described in Appendix E, in order to deal with higher-degree objective polynomials. Efficient derivation of approximate sparse certificates for high degree polynomials should allow to enlarge the spectrum of applicability of such techniques to larger size networks and broader classes of activation functions. This is an exciting topic of future research.

## Broader Impact

Developing optimization methods resulting in numerical certificates is a necessary step toward the verification of systems involving AI trained components. Such systems are expected to be more and more common in the transport industry and constitute a major challenge in terms of certification. We believe that polynomial optimization is one promising tool to address this challenge.

## Acknowledgments and Disclosure of Funding

This work has benefited from the AI Interdisciplinary Institute ANITI funding, through the French "Investing for the Future – PIA3" program under the Grant agreement n°ANR-19-PI3A-0004. Edouard Pauwels acknowledges the support of Air Force Office of Scientific Research, Air Force Material Command, USAF, under grant numbers FA9550-19-1-7026 and FA9550-18-1-0226, and ANR MaSDol. Victor Magron was supported by the FMJH Program PGMO (EPICS project) and EDF, Thales, Orange et Criteo, the Tremplin ERC Stg Grant ANR-18-ERC2-0004-01 (T-COPS project) as well as the European Union's Horizon 2020 research and innovation programme under the Marie Sklodowska-Curie Actions, grant agreement 813211 (POEMA). A large part of this work was carried out as Tong Chen was a master intern at IRT-Saint-Exupéry, Toulouse, France.

## Footnotes

[1]*Semidefinite programming (SDP)* is a subfield of convex conic optimization concerned with the optimization of a linear objective function over the intersection of the cone of positive semidefinite matrices with an affine subspace.

[2] https://openreview.net/forum?id=rJe4_xSFDB.

[3] https://www.gurobi.com/documentation/9.0/refman/crossover.html

[4]`https://worksheets.codalab.org/worksheets/0xa21e794020bb474d8804ec7bc0543f52/`

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
