[Supplementary Material]

# A Proof of Lemma 1

Denote by $D(x)$ the sub-differential of ReLU function, i.e. $D(x) = 1$ for $x > 0$, $D(x) = 0$ for $x < 0$ and $D(x) = [0, 1]$ for $x = 0$.

According to [5], $G$ has a closed graph and compact values. Furthermore, it holds that $G(t) \subseteq D(t)$ for all $t \in \mathbb{R}$. Adopting the terminology from [5], $D$ is *conservative* for the ReLU function, which implies that $G$ is conservative for the ReLU function as well [5, Remark 3(e)]. The formulation $G_{F_k}(\mathbf{x}_0) = (\prod_{i=1}^{m} \mathbf{A}_i^T \text{diag}(G(\mathbf{z}_i))) \mathbf{c}_k$ is an application of the chain rule of differentiation, where along each chain the conservative set-valued field $G$ is used in place of derivative for the ReLU function. By [5, Lemma 2], chain rule preserves conservativity, hence $G_{F_k}$ is a conservative mapping for function $F_k$. By conservativity [5], using convexity of $\Omega$ we have for all $\mathbf{x}, \mathbf{y} \in \Omega$, integrating along the segment.

$$
\begin{aligned}
|F_k(\mathbf{y}) - F_k(\mathbf{x})| &= \left| \int_{t=0}^{t=1} \max_{\mathbf{v} \in G_{F_k}(\mathbf{x}+t(\mathbf{y}-\mathbf{x}))} \langle \mathbf{y} - \mathbf{x}, \mathbf{v} \rangle \, dt \right| \\
&\leq \int_{t=0}^{t=1} \max_{\mathbf{v} \in G_{F_k}(\mathbf{x}+t(\mathbf{y}-\mathbf{x}))} \|\mathbf{y} - \mathbf{x}\| \|\mathbf{v}\|_* dt \\
&\leq \int_{t=0}^{t=1} \|\mathbf{y} - \mathbf{x}\| L_F^{||\cdot||} dt \\
&= L_F^{||\cdot||} \|\mathbf{y} - \mathbf{x}\|.
\end{aligned}
$$

*Remark (Being a gradient almost everywhere is not sufficient in general)*: As suggested by anonymous AC[5], Lemma 1 may admit a simpler proof. Indeed the Clarke subdifferential is the convex hull of limits of sequences of gradients. For Lipschitz constant, we want the maximum norm element, which necessarily happens at a corner of the convex hull, therefore for our purposes it suffices to consider sequences. Since the ReLU network will be almost-everywhere differentiable, we can consider a shrinking sequence of balls around any point, and we will have gradients which are arbitrarily close to any corner of the differential at our given point. Therefore, the norms will converge to that norm, and thus it suffices to optimize over differentiable points, and what we choose at the nondifferentiability does not matter.

The above argument is essentially valid because ReLU only contains univariate nondifferentiability which is very specific. However the argument is implicitly based on the idea that the composition of almost everywhere differentiable functions complies with calculus rules, which is not correct in general due to lack of injectivity. For more general networks, what we choose at the nondifferentiability does matter. Indeed, consider the following functions

$$
F: x \mapsto \begin{pmatrix} x \\ x \end{pmatrix}
$$

$$
G: \begin{pmatrix} y_1 \\ y_2 \end{pmatrix} \mapsto \max\{y_1, y_2\}
$$

The composition $G \circ F$ is the identity on $\mathbb{R}$ and both $F$ and $G$ are differentiable almost every where. Consider the mappings

$$
J_F: x \mapsto \begin{pmatrix} 1 \\ 1 \end{pmatrix}
$$

$$
J_G: \begin{pmatrix} y_1 \\ y_2 \end{pmatrix} \mapsto \begin{cases} (0,0)^T & \text{if } , y_1 = y_2 \\ (1,0)^T & \text{if } , y_1 > y_2 \\ (0,1)^T & \text{if } , y_1 < y_2 \end{cases}.
$$

$J_F$ is the Jacobian of $F$ and $J_G$ is the gradient of $G$ almost everywhere. Now the product

$$
J_G(F(x)) \times J_F(x) = 0
$$

for all $x \in \mathbb{R}$. Hence computing the product of these gradient almost everywhere operators gives value 0, suggesting that the function is constant, while it is not. The reason for the failure here is that

$J_G$, despite being gradient almost everywhere, is not conservative for $G$. For this reason, the product does not provide any notion of Lipschicity and the choice at nondifferentiability point does matter for $G$.

Overall, we believe that the proof suggested by AC is correct for ReLU networks because they are built only using univariate nondifferentiabilities. However the argument that, the choice made at nondifferentiability point does not have an impact, is not correct for more general networks. This discussion is kept explicit in this appendix since many other types of semialgebraic activation functions are used in deep learning, such as max pooling, for which one may use similar approaches as what we proposed to evaluate Lipschitz constants. For these more complex nondifferentiability, concervativity will be essential to ensure that one obtains proper Lipschitz constants, beyond the argument that beging differentiable almost everywhere implies that the choice made at nondifferentiability point does not matter.

# B  Illustration Examples of Lasserre's Hierarchy 3

## B.1  Dense Case 3.1

For illustration purpose and without going into details, consider the following simple example where we want to minimize $x_1 x_2$ over the unit disk on $\mathbb{R}^2$. That is:

$$\inf_{\mathbf{x} \in \mathbb{R}^2} \{f(\mathbf{x}) = x_1 x_2 : g(\mathbf{x}) = 1 - x_1^2 - x_2^2 \geq 0\}. \tag{5}$$

For $d = 1$, $\mathbf{y} = \{y_{00}, y_{01}, y_{10}, y_{20}, y_{11}, y_{02}\} \in \mathbb{R}^6$, $L_{\mathbf{y}}(f) = y_{11}$, and

$$\mathbf{M}_1(\mathbf{y}) = \begin{pmatrix} y_{00} & y_{10} & y_{01} \\ y_{10} & y_{20} & y_{11} \\ y_{01} & y_{11} & y_{02} \end{pmatrix}.$$

As $\omega = \lceil \deg(g)/2 \rceil = 1$, $M_0(g\mathbf{y}) \succeq 0$ simply translates to the linear constraint $L_{\mathbf{y}}(g) = 1 - y_{20} - y_{02} \geq 0$. Therefore (MomOpt-$d$) with $d = 1$ reads:

$$\inf_{\mathbf{y} \in \mathbb{R}^6} \{y_{11} : y_{00} = 1, \mathbf{M}_1(\mathbf{y}) \succeq 0, 1 - y_{20} - y_{02} \geq 0\}, \tag{6}$$

with optimal value $\rho_1 = -1/2 = f^*$. It turns out that (6) is exactly Shor's relaxation applied to (5). In fact, for QCQP the first-order moment relaxation (i.e., (MomOpt-$d$) with $d = 1$) is exactly Shor's relaxation.

## B.2  Sparse Case 3.2

For illustration, consider the following POP:

$$\inf_{\mathbf{x} \in \mathbb{R}^2} \{x_1 x_2 + x_2 x_3 : x_1^2 + x_2^2 \leq 1, x_2^2 + x_3^2 \leq 1\}. \tag{7}$$

Define the subsets $I_1 = \{1, 2\}$, $I_2 = \{2, 3\}$. It is easy to check that assumptions *A1*, *A2*, *A3* and *A4* hold. Define $\mathbf{y} = \{y_{000}, y_{100}, y_{010}, y_{001}, y_{200}, y_{110}, y_{101}, y_{020}, y_{011}, y_{002}\} \in \mathbb{R}^{10}$. For $d = 1$, the first-order dense moment matrix reads:

$$\mathbf{M}_1(\mathbf{y}) = \begin{pmatrix} y_{000} & y_{100} & y_{010} & y_{001} \\ y_{100} & y_{200} & y_{110} & y_{101} \\ y_{010} & y_{110} & y_{020} & y_{011} \\ y_{001} & y_{101} & y_{011} & y_{002} \end{pmatrix},$$

whereas the sparse moment matrix $\mathbf{M}_1(\mathbf{y}, I_1)$ (resp. $\mathbf{M}_1(\mathbf{y}, I_2)$) is the submatrix of $\mathbf{M}_1(\mathbf{y})$ taking red and pink (resp. blue and pink) entries. That is, $\mathbf{M}_1(\mathbf{y}, I_1)$ and $\mathbf{M}_1(\mathbf{y}, I_2)$ are submatrices of $\mathbf{M}_1(\mathbf{y})$, obtained by restricting to rows and columns concerned with subsets $I_1$ and $I_2$ only.

# C  Link between SDP and Sum-of-Square (SOS)

The primal and dual of Lasserre's hierarchy (MomOpt-$d$) nicely illustrate the duality between moments and positive polynomials. Indeed for each fixed $d$, the dual of (MomOpt-$d$) reads:

$$\sup_{t \in \mathbb{R}} \{t : f - t = \sigma_0 + \sum_{i=1}^{p} \sigma_i g_i\}, \tag{SOS-$d$}$$

where $\sigma_0$ is a *sum-of-squares (SOS)* polynomial of degree at most $2d$, and $\sigma_j$ are SOS polynomials of degree at most $2(d - \omega_i)$, $\omega_i = \lceil \deg(g_j)/2 \rceil$. The right-hand-side of the identity in (SOS-$d$) is nothing less than Putinar's positivity certificate [29] for the polynomial $\mathbf{x} \mapsto f(\mathbf{x}) - t$ on the compact semialgebraic set $\{\mathbf{x} : g_i(\mathbf{x}) \geq 0, \ i \in [p]\}$.

Similarly, the dual problem of (MomSpOpt-$d$) reads:

$$\sup_{t \in \mathbb{R}} \{ t : f - t = \sum_{k=1}^{l} \left( \sigma_{0,k} + \sum_{j=1}^{m} \sigma_{j,k} g_j \right) \}, \qquad \text{(SpSOS-}d\text{)}$$

where $\sigma_{0,k}$ are SOS in $\mathbb{R}[\mathbf{x}_{I_k}]$ of degree at most $2d$, and $\sigma_{j,k}$ are SOS in $\mathbb{R}[\mathbf{x}_{I_k}]$ of degree at most $2(d - \omega_i)$, $\omega_i = \lceil \deg(g_j)/2 \rceil$. Then (SpSOS-$d$) implements the sparse Putinar's positivity certificate [18, 37].

## D    Illustration of Heuristic Relaxation

Consider problem (NlySpOpt). We already have a sparsity pattern with subsets $I_k$ and an additional "bad" constraint $g \geq 0$ (assumed to be quadratic). Then we consider the sparse moment relaxations (MomSpOpt-$d$) applied to (NlySpOpt) *without* the bad constraint $g \geq 0$ and simply add two constraints: (i) the moment constraint $\mathbf{M}_1(\mathbf{y}) \succeq 0$ (with full dense first-order moment matrix $\mathbf{M}_1(\mathbf{y})$), and (ii) the linear moment inequality constraint $L_{\mathbf{y}}(g) \geq 0$ (which is the lowest-order localizing matrix constraint $\mathbf{M}_0(g\,\mathbf{y}) \succeq 0$).

To see why the full moment constraint $\mathbf{M}_1(\mathbf{y}) \succeq 0$ is needed, consider the toy problem (7). Recall that the subsets we defined are $I_1 = \{1, 2\}$, $I_2 = \{2, 3\}$. Now suppose that we need to consider an additional "bad" constraint $(1 - x_1 - x_2 - x_3)^2 = 0$. After developing $L_{\mathbf{y}}(g)$, one needs to consider the moment variable $y_{103}$ corresponding to the monomial $x_1 x_3$ in the expansion of $g = (1 - x_1 - x_2 - x_3)^2$, and $y_{103}$ does *not* appear in the moment matrices $\mathbf{M}_d(\mathbf{y}, I_1)$ and $\mathbf{M}_d(\mathbf{y}, I_2)$ because $x_1$ and $x_3$ are not in the same subset. However $y_{103}$ appears in $\mathbf{M}_1(\mathbf{y})$ (which is a $n \times n$ matrix).

Now let us see how this works for problem (LCEP). First introduce new variables $\mathbf{z}_i$ with associated constraints $\mathbf{z}_i - \mathbf{A}_i \mathbf{x}_{i-1} - \mathbf{b}_i = 0$, so that all "bad" constraints are affine. Equivalently, we may and will consider the single "bad" constraint $g \geq 0$ with $g(\mathbf{z}_1, \ldots, \mathbf{x}_0, \mathbf{x}_1, \ldots) = -\sum_i \|\mathbf{z}_i - \mathbf{A}\mathbf{x}_{i-1} - \mathbf{b}_i\|^2$ and solve (MomNlySpOpt-$d$). We briefly sketch the rationale behind this reformulation. Let $(\mathbf{y}^d)_{d \in \mathbb{N}}$ be a sequence of optimal solutions of (MomNlySpOpt-$d$). If $d \to \infty$, then $\mathbf{y}^d \to \mathbf{y}$ (possibly for a subsequence $(d_k)_{k \in \mathbb{N}}$), and $\mathbf{y}$ corresponds to the moment sequence of a measure $\mu$, supported on $\{(\mathbf{x}, \mathbf{z}) : g_i(\mathbf{x}, \mathbf{z}) \geq 0, \ i \in [p]; \int g \, d\mu \geq 0\}$. But as $-g$ is a square, $\int g d\mu \geq 0$ implies $g = 0$, $\mu$-a.e., and therefore $\mathbf{z}_i = \mathbf{A}\mathbf{x}_{i-1} + \mathbf{b}_i$, $\mu$-a.e. This is why we do not need to consider the higher-order constraints $\mathbf{M}_d(g\,\mathbf{y}) \succeq 0$ for $d > 0$; only $\mathbf{M}_0(g\,\mathbf{y}) \succeq 0$ ($\Leftrightarrow L_{\mathbf{y}}(g) \geq 0$) suffices. In fact, we impose the stronger linear constraints $L_{\mathbf{y}}(g) = 0$ and $L_{\mathbf{y}}(\mathbf{z}_i - \mathbf{A}\mathbf{x}_{i-1} - \mathbf{b}_i) = 0$ for all $i \in [p]$.

## E    Lifting and Approximation Techniques for Cubic Terms

As discussed at the end of Section 4, for 2-hidden layer networks, one needs to reduce the objective function to degree 2 so that the **HR-2** algorithm can be adapted to problem (LCEP). Precisely, problem (LCEP) for 2-hidden layer networks is the following POP:

$$\max_{\mathbf{x}_i, \mathbf{u}_i, \mathbf{t}} \quad \mathbf{t}^T \mathbf{A}_1^T \mathrm{diag}(\mathbf{u}_1) \mathbf{A}_2^T \mathrm{diag}(\mathbf{u}_2) \mathbf{c} \qquad \text{(LCEP-MLP}_2\text{)}$$

$$\text{s.t.} \quad \begin{cases} \mathbf{u}_1(\mathbf{u}_1 - 1) = 0, (\mathbf{u}_1 - 1/2)(\mathbf{A}_1 \mathbf{x}_0 + \mathbf{b}_1) \geq 0, \\ \mathbf{u}_2(\mathbf{u}_2 - 1) = 0, (\mathbf{u}_2 - 1/2)(\mathbf{A}_2 \mathbf{x}_1 + \mathbf{b}_2) \geq 0, \\ \mathbf{x}_1(\mathbf{x}_1 - \mathbf{A}_1 \mathbf{x}_0 - \mathbf{b}_1) = 0, \mathbf{x}_1 \geq 0, \mathbf{x}_1 \geq \mathbf{A}_1 \mathbf{x}_0 + \mathbf{b}_1 ; \\ \mathbf{t}^2 \leq 1, (\mathbf{x}_0 - \bar{\mathbf{x}}_0 + \varepsilon)(\mathbf{x}_0 - \bar{\mathbf{x}}_0 - \varepsilon) \leq 0 . \end{cases}$$

## E.1 Lifting Technique

Define new decision variable $\mathbf{s} := \mathbf{u}_1 \mathbf{u}_2^T$, so that the degree of objective is reduced to 2. Problem (LCEP-MLP$_2$) can now be reformulated as:

$$\max_{\mathbf{x}_i, \mathbf{u}_i, \mathbf{t}} \quad \sum_i t_i \langle \mathrm{diag}(\mathbf{A}_1^{(:,i)}) \mathbf{A}_2^T \mathrm{diag}(\mathbf{c}), \mathbf{s} \rangle \qquad \text{(ReducedLCEP-MLP}_2\text{)}$$

$$\text{s.t.} \quad \begin{cases} \mathbf{u}_1(\mathbf{u}_1 - 1) = 0, (\mathbf{u}_1 - 1/2)(\mathbf{A}_1 \mathbf{x}_0 + \mathbf{b}_1) \geq 0, \\ \mathbf{u}_2(\mathbf{u}_2 - 1) = 0, (\mathbf{u}_2 - 1/2)(\mathbf{A}_2 \mathbf{x}_1 + \mathbf{b}_2) \geq 0, \\ \mathbf{x}_1(\mathbf{x}_1 - \mathbf{A}_1 \mathbf{x}_0 - \mathbf{b}_1) = 0, \mathbf{x}_1 \geq 0, \mathbf{x}_1 \geq \mathbf{A}_1 \mathbf{x}_0 + \mathbf{b}_1; \\ \mathbf{t}^2 \leq 1, (\mathbf{x}_0 - \bar{\mathbf{x}}_0 + \varepsilon)(\mathbf{x}_0 - \bar{\mathbf{x}}_0 - \varepsilon) \leq 0, \mathbf{s} = \mathbf{u}_1 \mathbf{u}_2^T. \end{cases}$$

For (ReducedLCEP-MLP$_2$), we have $p_1 p_2$ more variables ($\mathbf{s}$) and constraints ($\mathbf{s} = \mathbf{u}_1 \mathbf{u}_2^T$), where $\mathbf{u}_1 \in \mathbb{R}^{p_1}$ and $\mathbf{u}_2 \in \mathbb{R}^{p_2}$. Even when $p_1 = p_2 = 100$, we add 10000 variables and constraints, which will cause a memory issue (no SDP solver is able to handle matrices of size $O(10^4)$). This is why we use the following approximation technique as a remedy.

## E.2 Heuristic Relaxation for Cubic Terms

In this section, we introduce an alternative technique to handle the cubic terms $t^i u_1^j u_2^k$ appearing in the objective function of problem (LCEP-MLP$_2$). Recall that the main obstacle that prevents us from applying the **HR-2** method is that we don't have the moments for cubic terms $t^i u_1^j u_2^k$ in the first-order moment matrix $\mathbf{M}_1(\mathbf{y}, \{t^i, u_1^j, u_2^k\})$. Precisely, we only have the moments of quadratic terms in $\mathbf{M}_1(\mathbf{y}, \{t^i, u_1^j, u_2^k\})$:

$$\mathbf{M}_1(\mathbf{y}, \{t^i, u_1^j, u_2^k\}) = \begin{pmatrix} L_{\mathbf{y}}(1) & L_{\mathbf{y}}(t^i) & L_{\mathbf{y}}(u_1^j) & L_{\mathbf{y}}(u_2^k) \\ L_{\mathbf{y}}(t^i) & L_{\mathbf{y}}((t^i)^2) & L_{\mathbf{y}}(t^i u_1^j) & L_{\mathbf{y}}(t^i u_2^k) \\ L_{\mathbf{y}}(u_1^j) & L_{\mathbf{y}}(u_1^j t^i) & L_{\mathbf{y}}((u_1^j)^2) & L_{\mathbf{y}}(u_1^j u_2^k) \\ L_{\mathbf{y}}(u_2^k) & L_{\mathbf{y}}(u_2^k t^i) & L_{\mathbf{y}}(u_2^k u_1^j) & L_{\mathbf{y}}((u_2^k)^2) \end{pmatrix}$$

The moments of cubic terms $t^i u_1^j u_2^k$ lie in the second-order moment matrix $\mathbf{M}_2(\mathbf{y}, \{t^i, u_1^j, u_2^k\})$, which is of size $\binom{3+2}{2} = 10$. However, since we only need the moments of the cubic terms, a submatrix of $\mathbf{M}_2(\mathbf{y})$ suffices:

$$\mathbf{M}_2^{sub}(\mathbf{y}, \{t^i, u_1^j, u_2^k\}) = \begin{pmatrix} L_{\mathbf{y}}(1) & L_{\mathbf{y}}(t^i) & L_{\mathbf{y}}(u_1^j u_2^k) \\ L_{\mathbf{y}}(t^i) & L_{\mathbf{y}}((t^i)^2) & L_{\mathbf{y}}(t^i u_1^j u_2^k) \\ L_{\mathbf{y}}(u_1^j u_2^k) & L_{\mathbf{y}}(t^i u_1^j u_2^k) & L_{\mathbf{y}}((u_1^j)^2 (u_2^k)^2) \end{pmatrix}$$

Thus, in order to obtain the moments of cubic terms, one only needs to put $\mathbf{M}_1(\mathbf{y})$ and $\mathbf{M}_2^{sub}(\mathbf{y}, \{t^i, u_1^j, u_2^k\})$ for each cubic term $t^i u_1^j u_2^k$ together. Recall that for problem (LCEP-MLP$_2$), $\mathbf{t} \in \mathbb{R}^{p_0}$, $\mathbf{u}_1 \in \mathbb{R}^{p_1}$, $\mathbf{u}_2 \in \mathbb{R}^{p_2}$. Define the subsets for (LCEP-MLP$_2$) as $I^i = \{x_0^i, t^i\}$ for $i \in [p_0]$; $J_1^j = \{x_1^j, z_1^j\}, J_2^j = \{u_1^j, z_1^j\}$ for $j \in [p_1]$; $K^k = \{u_2^k, z_2^k\}$ for $k \in [p_2]$. Then the second-order

heuristic relaxation (**HR-2**) for problem (LCEP-MLP$_2$) reads as:

$$\sup_{\mathbf{y}}\{L_{\mathbf{y}}(\mathbf{t}^T\mathbf{A}_1^T\mathrm{diag}(\mathbf{u}_1)\mathbf{A}_2^T\mathrm{diag}(\mathbf{u}_2)\mathbf{c}) : L_{\mathbf{y}}(1) = 1, \mathbf{M}_1(\mathbf{y}) \succeq 0\,;$$

$$\mathbf{M}_2^{sub}(\mathbf{y}, \{t^i, u_1^j, u_2^k\}) \succeq 0, i \in [p_0], j \in [p_1], k \in [p_2]\,;$$

$$L_{\mathbf{y}}(\mathbf{z}_1 - \mathbf{A}_1\mathbf{x}_0 - \mathbf{b}_1) = 0, L_{\mathbf{y}}((\mathbf{z}_1 - \mathbf{A}_1\mathbf{x}_0 - \mathbf{b}_1)^2) = 0\,;$$

$$L_{\mathbf{y}}(\mathbf{z}_2 - \mathbf{A}_2\mathbf{x}_1 - \mathbf{b}_2) = 0, L_{\mathbf{y}}((\mathbf{z}_2 - \mathbf{A}_2\mathbf{x}_1 - \mathbf{b}_2)^2) = 0\,;$$

$$\mathbf{M}_2(\mathbf{y}, J_2^j) \succeq 0, \mathbf{M}_1(u_1^j(u_1^j - 1)\mathbf{y}, J_2^j) = 0, \mathbf{M}_1((u_1^j - 1/2)z_1^j\mathbf{y}, J_2^j) \succeq 0, j \in [p_1]\,;$$

$$\mathbf{M}_2(\mathbf{y}, K^k) \succeq 0, \mathbf{M}_1(u_2^k(u_2^k - 1)\mathbf{y}, K^k) = 0, \mathbf{M}_1((u_2^k - 1/2)z_2^k\mathbf{y}, K^k) \succeq 0, k \in [p_2]\,;$$

$$\mathbf{M}_2(\mathbf{y}, J_1^j) \succeq 0, \mathbf{M}_1(x_1^j(x_1^j - z_1^j)\mathbf{y}, J_1^j) = 0,$$

$$\mathbf{M}_1(x_1^j\mathbf{y}, J_1^j) \succeq 0, \mathbf{M}_1((x_1^j - z_1^j)\mathbf{y}, J_1^j) \succeq 0, j \in [p_1]\,;$$

$$\mathbf{M}_2(\mathbf{y}, I_1^i) \succeq 0, \mathbf{M}_1((1 - (t^i)^2)\mathbf{y}, I^i) \succeq 0,$$

$$\mathbf{M}_1(-(x_0^i - \bar{x}_0^i + \varepsilon)(x_0^i - \bar{x}_0^i - \varepsilon)\mathbf{y}, I^i) \succeq 0, i \in [p_0]\}\,.$$
$$\text{(MomLCEP}_2\text{-2)}$$

In this way, we add $p_0p_1p_2$ moment matrices $\mathbf{M}_2^{sub}(\mathbf{y}, \{t^i, u_1^j, u_2^k\})$ of size 3, and $p_0p_1p_2 + p_1p_2$ moment variables $L_{\mathbf{y}}(t^iu_1^ju_2^k)$, $L_{\mathbf{y}}((u_1^j)^2(u_2^k)^2)$. A variant of this technique is to enlarge the size of the moment matrices but in the meantime reduce the number of moment matrices. For instance, consider the following submatrix of the second-order moment matrix $\mathbf{M}_2(\mathbf{y}, \{\mathbf{t}, \mathbf{y}_1, u_2^k\})$:

$$\mathbf{M}_2^{sub}(\mathbf{y}, \{\mathbf{t}, \mathbf{u}_1, u_2^k\}) = \begin{pmatrix} L_{\mathbf{y}}(1) & L_{\mathbf{y}}(\mathbf{t}^T) & L_{\mathbf{y}}(\mathbf{u}_1^Tu_2^k) \\ L_{\mathbf{y}}(\mathbf{t}) & L_{\mathbf{y}}(\mathbf{t}\mathbf{t}^T) & L_{\mathbf{y}}(\mathbf{t}\mathbf{u}_1^Tu_2^k) \\ L_{\mathbf{y}}(\mathbf{u}_1u_2^k) & L_{\mathbf{y}}(\mathbf{u}_1\mathbf{t}^Tu_2^k) & L_{\mathbf{y}}(\mathbf{u}_1\mathbf{u}_1^T(u_2^k)^2) \end{pmatrix} \qquad (8)$$

We have all the moments of the cubic terms $t^iu_1^ju_2^k$ from those $\mathbf{M}_2^{sub}(\mathbf{y}, \{\mathbf{t}, \mathbf{u}_1, u_2^k\})$. However, in this case, we only add $p_2$ moment matrices $\mathbf{M}_2^{sub}(\mathbf{y}, \{\mathbf{t}, \mathbf{u}_1, u_2^k\})$ of size $1 + p_0 + p_1$, and $p_0p_1p_2 + p_1^2p_2$ new variables $L_{\mathbf{y}}(\mathbf{u}_1\mathbf{t}^Tu_2^k)$, $L_{\mathbf{y}}(\mathbf{u}_1\mathbf{u}_1^T(u_2^k)^2)$. Note that we can also use the first-order heuristic relaxation (**HR-1**), which is formulated as:

$$\sup_{\mathbf{y}}\{L_{\mathbf{y}}(\mathbf{t}^T\mathbf{A}_1^T\mathrm{diag}(\mathbf{u}_1)\mathbf{A}_2^T\mathrm{diag}(\mathbf{u}_2)\mathbf{c}) : L_{\mathbf{y}}(1) = 1, \mathbf{M}_1(\mathbf{y}) \succeq 0,$$

$$\mathbf{M}_2^{sub}(\mathbf{y}, \{t^i, u_1^j, u_2^k\}) \succeq 0, i \in [p_0], j \in [p_1], k \in [p_2]\,,$$

$$L_{\mathbf{y}}(\mathbf{u}_1(\mathbf{u}_1 - 1)) = 0, L_{\mathbf{y}}((\mathbf{u}_1 - 1/2)\mathbf{z}_1) \geq 0\,,$$

$$L_{\mathbf{y}}(\mathbf{u}_2(\mathbf{u}_2 - 1)) = 0, L_{\mathbf{y}}((\mathbf{u}_2 - 1/2)\mathbf{z}_2) \geq 0\,,$$

$$L_{\mathbf{y}}(\mathbf{x}_1(\mathbf{x}_1 - \mathbf{z}_1)) = 0, L_{\mathbf{y}}(\mathbf{x}_1) \geq 0, L_{\mathbf{y}}(\mathbf{x}_1 - \mathbf{z}_1) \geq 0\,;$$

$$L_{\mathbf{y}}(\mathbf{t}^2 - 1) \leq 0, L_{\mathbf{y}}((\mathbf{x}_0 - \bar{\mathbf{x}}_0 + \varepsilon)(\mathbf{x}_0 - \bar{\mathbf{x}}_0 - \varepsilon)) \leq 0\}\,. \qquad \text{(MomLCEP}_2\text{-1)}$$

# F   Global Lipschitz Constant Estimation for Random Networks

We use the experimental settings described in Section 5.

## F.1   1-Hidden Layer Networks

Figure 2 displays the average upper bounds of global Lipschitz constants and the *t*ime of different algorithms for 1-hidden layer random networks of different sizes and sparsities. We can see from Figure 2a that when the size of the network is small (10, 20, etc.), the LP-based method **LipOpt-3** is slightly better than the SDP-based method **HR-2**. However, when the size and sparsity of the network increase, **HR-2** provides tighter bounds. From Figure 2b, we can see that **LipOpt-3** is more efficient than **HR-2** only when the size or the sparsity of the network is small (for $(10, 10)$ networks, or for $(40, 40)$ networks of sparsity 5, etc.). When the networks are dense or nearly dense, our method not only takes much less time, but also gives much tighter upper bounds. For global Lipschitz constant estimation, **SHOR** and **HR-2** give nearly the same upper bounds. This is because the sizes of the toy networks are quite small. For big real network, as shown in Table 2, **HR-2** provides strictly tighter bound than **SHOR**. Finally, **SHOR** is more efficient than **HR-2** and **LipOpt-3** in terms of computational complexity.

Figure 2: Global Lipschitz constant upper bounds (left) and solver running time (right) for 1-hidden layer networks with respect to $L_\infty$-norm obtained by **SHOR**, **HR-2**, **LipOpt-3**, **LipOpt-4** and **LBS**. We generate random networks of size 10, 20, 40, 80. For size 10, we consider sparsity 4, 8, 12, 16, 20; for size 20, we consider sparsity 8, 16, 24, 32, 40; for size 40 and 80, we consider sparsity 10, 20, 30, 40, 50, 60, 70, 80. In the meantime, we display median and quartiles over 10 random networks draws.

### F.2   2-Hidden Layer Networks

For 2-hidden layer networks, we use the technique introduced in Appendix E in order to deal with the cubic terms in the objective. Figure 3 displays the average upper bounds of global Lipschitz constants and the running time of different algorithms for 2-hidden layer random networks of different sizes and sparsities. We can see from Figure 3a that the SDP-based method **HR-2** performs worse than the LP-based method **LipOpt-3** for networks of size $(10, 10, 10)$. However, as the size and the sparsity of the network increase, the difference between **HR-2** and **LipOpt-3** becomes smaller (and **HR-2** performs even better). For networks of size $(20, 20, 10)$, $(30, 30, 10)$ and $(40, 40, 10)$, with sparsity greater than 10, **HR-2** provides strictly tighter bounds than **LipOpt-3**. This fact has already been shown in Table 1 (right), **HR-1** and **HR-2** give consistently tighter upper bounds than **LipOpt-3**, with the price of higher computational time.

## G   Local Lipschitz Constant Estimation for Random Networks

We use the experimental settings described in Section 5.

### G.1   1-Hidden Layer Networks

Figure 4 displays the average upper bounds of local Lipschitz constants and the running time of different algorithms for 1-hidden layer random networks of different sizes and sparsities. By contrast with the global case, we can see from Figure 4a that **HR-2** gives strictly tighter upper bounds than **SHOR** when the sparsity is larger than 40. As a trade-off, **HR-2** takes more computational time than **SHOR**. According to Figure 4b, the running time of **HR-2** is around 5 times longer than **SHOR**. Similar to the global case, **LipOpt-3** performs well when the network is sparse. However, when the sparsity increases, **HR-2** and **SHOR** are more efficient and provide better bounds than **LipOpt-3**.

### G.2   2-Hidden Layer Networks

For 2-hidden layer networks, we use the approximation technique described in Appendix E in order to reduce the objective to degree 2. Figure 5a and 5b displays the average upper bounds of local Lipschitz constants and the running time of different algorithms for 2-hidden layer random networks of different sizes and sparsities. By contrast with the global case, we can see from Figure 5a that **HR-2** gives strictly tighter upper bounds than **HR-1** when the sparsity is larger than 40. As a trade-off,

Figure 3: Global Lipschitz constant upper bounds (left) and solver running time (right) for 2-hidden layer networks with respect to $L_\infty$-norm obtained by **HR-2**, **HR-1**, **LipOpt-3**, **LipOpt-4** and **LBS**. We generate random networks of size 10, 20, 30, 40. For size $(10, 10, 10)$, we consider sparsity 4, 8, 12, 16, 20; for size $(20, 20, 10)$, we consider sparsity 4, 8, 12, 16, 20, 24, 28, 32, 36, 40; for size $(30, 30, 10)$, we consider sparsity 10, 20, 30, 40, 50, 60; for size $(40, 40, 10)$, we consider sparsity 10, 20, 30, 40, 50, 60, 70, 80. In the meantime, we display median and quartiles over 10 random networks draws.

Figure 4: Local Lipschitz constant upper bounds (left) and solver running time (right) for 1-hidden layer networks with respect to $L_\infty$-norm obtained by **HR-2**, **SHOR**, **LipOpt-3** and **LBS**. By default, $\varepsilon = 0.1$. We generate random networks of size 10, 20, 40, 80. For size 10, we consider sparsity 4, 8, 12, 16, 20; for size 20, we consider sparsity 8, 16, 24, 32, 40; for size 40 and 80, we consider sparsity 10, 20, 30, 40, 50, 60, 70, 80. In the meantime, we display median and quartiles over 10 random networks draws.

**HR-2** takes more computational time than **HR-1**. According to Figure 5b, the running time of **HR-2** is just around 3 times longer than **HR-1**. Similar to the global case, **LipOpt-3** performs well when the network is sparse. However, when the sparsity increases, **HR-2** and **HR-1** provide better bounds than **LipOpt-3**, with the price of higher computational time..

(a)  (b)

Figure 5: Local Lipschitz constant upper bounds (left) and solver running time (right) for 2-hidden layer networks with respect to $L_\infty$-norm obtained by **HR-2**, **HR-1**, **LipOpt-3** and **LBS**. By default, $\varepsilon = 0.1$. We generate random networks of size 20, 30, 40, 50. For size $(10, 10, 10)$, we consider sparsity 4, 8, 12, 16, 20; for size $(20, 20, 10)$, we consider sparsity 4, 8, 12, 16, 20, 24, 28, 32, 36, 40; for size $(30, 30, 10)$, we consider sparsity 10, 20, 30, 40, 50, 60; for size $(40, 40, 10)$, we consider sparsity 10, 20, 30, 40, 50, 60, 70, 80. In the meantime, we display median and quartiles over 10 random networks draws.

# H   Robustness Certification for MNIST Network SDP-NN [30]

The matrix of Lipschitz constants of function $f_{i,j}$ (defined in the second paragraph of Section 5.2) with respect to input $\mathbf{x}_0 = \mathbf{0}$, $\epsilon = 2$ and norm $\|\cdot\|_\infty$:

$$
\mathbf{L} = \begin{pmatrix}
* & 7.94 & 7.89 & 8.28 & 8.64 & 8.10 & 7.66 & 8.04 & 7.46 & 8.14 \\
7.94 & * & 7.74 & 7.36 & 7.68 & 8.81 & 8.06 & 7.55 & 7.36 & 8.66 \\
7.89 & 7.74 & * & 7.63 & 8.81 & 10.23 & 8.18 & 8.13 & 7.74 & 9.08 \\
8.28 & 7.36 & 7.63 & * & 8.52 & 7.74 & 9.47 & 8.01 & 7.37 & 7.96 \\
8.64 & 7.68 & 8.81 & 8.52 & * & 9.44 & 7.98 & 8.65 & 8.49 & 7.47 \\
8.10 & 8.81 & 10.23 & 7.74 & 9.44 & * & 8.26 & 9.26 & 8.17 & 8.55 \\
7.66 & 8.06 & 8.18 & 9.47 & 7.98 & 8.26 & * & 10.18 & 8.00 & 9.83 \\
8.04 & 7.55 & 8.13 & 8.01 & 8.65 & 9.26 & 10.18 & * & 8.28 & 7.65 \\
7.46 & 7.36 & 7.74 & 7.37 & 8.49 & 8.17 & 8.00 & 8.28 & * & 7.87 \\
8.14 & 8.66 & 9.08 & 7.96 & 7.47 & 8.55 & 9.83 & 7.65 & 7.87 & *
\end{pmatrix}
$$

where $\mathbf{L} = (L_{ij})_{i \neq j}$. Note that if we replace the vector $\mathbf{c}$ in (LCEP) by $-\mathbf{c}$, the problem is equivalent to the original one. Therefore, the matrix $\mathbf{L}$ is symmetric, and we only need to compute 45 Lipschitz constants (the upper triangle of $\mathbf{L}$).

Figure 6 shows several certified and non-certified examples taken from the MNIST test dataset.

Figure 6: Examples of certified points (above) and non-certified points (bellow).

## Footnotes

[5]This is the occasion to thank this person for his/her work on the paper, and for this nice suggestion.