[Reviews · NeurIPS 2020]

Review 1

Summary and Contributions: ***Update after rebuttal*** I have read authors response. My opinion remains that this is a good paper and I recommend acceptance. Even though Lemma 1 can be derived without much problem from ref [5], It provides crucial theoretical grounding to the paper. As far as I know, all applications of such bounds rely precisely on their tightness, so I don't think comparison of the methods in robustness certification (or other) applications would necessarily complement the paper. This type of "verification" problems are much harder than training, so in my opinion it is ok to see experiments with smaller networks than used in industry. *** The papers presents a hierarchical optimization approach, based on polynomial optimization, to compute an upper bound on the Lipschtiz constant of ReLU networks. The method is computationally expensive and so heuristics are employed to provide algorithms that are able to work with larger instances of networks, while always providing a valid upper bound on the Lipschtiz constant.

Strengths: The semialgebraic description of the ReLU and its derivative is a tighter description than the ones appearing in previous work, thus they can potentially provide tighter bounds. Precisely, one of the main strengths of this paper is that it builds upon previous work, which introduced a polynomial optimization approach to provide upper bounds on Lipschitz constants. Thus, it has strong theoretical roots and many more improvements might be derived in the future. It is also remarkable that the non-smoothness is properly handled, and in particular, the issues with the incorrectness of backpropagation on ReLU networks is correctly addressed, thus filling a theoretical hole usually present and overlooked in many other works. As another strength, on the practical side the implementation of the methods seem to be more efficient in their use of memory and computing power, as the reported times and memory consumption, compared to the baselines. are much better. It would be a good idea to publicly share the code for the community to use.

Weaknesses: One of the main weaknesses is the lack of comparison with the local lispchitz constant estimation baseline. If the code for the baseline was not available perhaps it could have been implemented easily?

Correctness: The claims are properly supported with strong theory, I have checked proof of lemma 1 in appendix. The empirical methodology seems ok.

Clarity: The paper writing is ok but some parts might benefit from improving readability. In particular the introduction seems a bit lacking in style. Also the abstract and title might need slight changes in wording. Also there is some debatable claim "However this method is restricted to the L2-norm whereas most deep learning applications are rather concerned with the Lāˆž-norm." : the prevalence of the L_infinity norm is found in the robust-deep-learning literature. I dont think that it is ok to claim that such norm is 'globally' more important than the l2 norm in deep learning applications.

Relation to Prior Work: The paper compares to previous work that follows the same polynomial optimization-based approach, while also mentioning and referencing other type of algorithms for this task. The differences are clearly stated.

Reproducibility: Yes

Additional Feedback:


Review 2

Summary and Contributions: This paper writes the ReLU and its generalized derivative in terms of polynomial inequalities, using this to write down a polynomial optimization problem whose value exactly equals the Lipschitz constant of a ReLU network, and gives a way of approximately solving it using a relaxation of Lasserre's hierarchy.

Strengths: The underlying idea is very nice (expressing the ReLU and its generalized derivative with polynomial inequalities, and utilizing this in the optimization problem calculating the network's Lipschitz constant). I could see this being an important technique for analyzing ReLU networks, and as a core contribution is very deserving of praise. Also, it is noteworthy that the authors showed this formulation could be exploited by via a relaxation of the Lasserre/SDP hierarchy. The results compare quite favorably to reference 21, an LP relaxation for computing the Lipschitz constant.

Weaknesses: It's difficult to see how this would work for modern neural nets, because the authors don't really go into detail on what happens with more layers, and how well the heuristic relaxation fares in that case (it looks like it might not fare that well because the relaxation appears to be dropping a lot of things with each layer). Though, to be fair, it seems the previous LP method (which was accepted to ICLR) seems to suffer the same issue of scaling deeper, but because of computational infeasibility. Also, from what I can tell the relaxation doesn't necessarily provide a valid upper bound (a problem from the SDP hierarchy might not, and a relaxation of it is even less likely to) -- wish this was commented on more. *Edit* -- my above remark is wrong, yes their method provides a valid upper bound because it is a relaxation of a maximization problem.

Correctness: The application of the reference generalizing differentiation to Lemma 1 was nice. Overall, the application of the Lasserre hierarchy and the relaxation used make sense.

Clarity: As someone not too familiar with the area, I initially found the paper hard to understand, and had to read some of the textbook in the references to get a basic grasp on what was going on. Perhaps the paper is quite clear to someone with more experience with polynomial optimization, though. Actually, the writing is quite efficient, and I don't think there are any major omissions of information, but perhaps the paper could be written a bit friendlier. Suggestions for clarity: - When describing table 1 in lines 89-94, make it clear whose methods are what. - To make the paper more self-contained, if there is not space in the main paper, I would suggest explaining the features of MomOpt-d i.e. what are the moment matrices, how is this a relaxation, etc. Or, to be more specific in what parts of the polynomial optimization textbook are relevant. - line 194, I think x0 should be x_0? - For MomLCEP-2, why not just have the constraint Ax+b=z? - I think more context-providing statements would be good in general, e.g. when introducing Lasserre hierarchy, say you will use a relaxation of it to approximately solve LCEP, and are just giving an overview; more commentary on why you are considering 1-hidden layer in the main paper (in particular extending to multiple layers seems hacky, which would seem to make applying this to W-GAN difficult)

Relation to Prior Work: The inclusion of the semialgebraic constraints determining the ReLU operation and its generalized derivative is a clear improvement upon the optimization problem posed by the paper this paper compares itself to. It's great that this paper provides a unique way of specifically addressing networks that use the ReLU, which is an important activation function. Furthermore, it makes sense why their method (with the heuristic relaxation) would be much faster than the one with the LP.

Reproducibility: Yes

Additional Feedback: * I reduced my score from 8 to 7 because of issues other reviewers brought up about lacking practicality; I agree with them that it would be good to show an application of the improved Lipschitz constant estimation on a downstream task. Also, perhaps the relaxation of the sparse structural constraints could be explained more clearly, and perhaps writing out a full example, or comparing numbers of variables/constraints of this reference and others, etc., could aid clarity. However, I still think the exact formulation of the Lipschitz constant is very nice.


Review 3

Summary and Contributions: This paper proposes a semidefinite programming hierarchy heuristic approach to compute the upper bounds on the Lipschitz constant of neural networks. In deriving this heuristic, the paper also proposes the semialgebraic expression of the generalized derivative of ReLU function. Empirical results show that the proposed method provides a trade-off between better bounds and the solver running time compared to the state-of-the-art methods. This is a mediocre paper, and I am leaning to reject it because (1) there is a lack of experimental results on robustness certification or other applications that need the estimation of the Lipschitz constant of neural networks, and (2) even the experimental results provided in the paper do not show a clear advantage of the proposed method compared to other baseline methods.

Strengths: The paper addresses an interesting topic of estimating the Lipschitz constant of a neural network, which is then useful for robustness certification. The background work and the proposed method are clearly presented.

Weaknesses: First, other works in this area usually show the advantage of the estimated Lipschitz constant on robustness certification tasks or other downstream tasks, empirically or theoretically (see [1], [2], and [3]). Without those experiments and only looking at the upper bounds of the Lipschitz constant of neural networks, it is hard to evaluate the significance of the work. Second, the upper bounds of the Lipschitz constant and the corresponding computational times also do not clearly show the advantage of the proposed method. For example, in table 2, when computing upper bounds of global Lipschitz constant and the solver running time on the trained network, while the proposed method achieves better upper bounds, it requires much more computational time. Again, this further emphasizes the need of additional experiments on downstream tasks such as robustness certification to justify the significance of the method. [1] Aditi Raghunathan, Jacob Steinhardt, and Percy Liang. Semidefinite relaxations for certifying robustness to adversarial examples. In Advances in Neural Information Processing Systems, pages 10877ā€“10887, 2018. [2] Yusuke Tsuzuku, Issei Sato, and Masashi Sugiyama. Lipschitz-margin training: Scalable certification of perturbation invariance for deep neural networks. In Advances in Neural Information Processing Systems, pages 6541ā€“6550, 2018. [3] Krishnamurthy Dvijotham, Robert Stanforth, Sven Gowal, Timothy A Mann, and Pushmeet Kohli. A dual approach to scalable verification of deep networks. In UAI, pages 550ā€“559, 2018.

Correctness: The claims and method are correct. Additional experiments are needed.

Clarity: The paper is well-written.

Relation to Prior Work: It is clearly discussed how the proposed method differs from previous works.

Reproducibility: Yes

Additional Feedback: I am willing to increase my score if the authors provide experiments on robustness certification or other downstream tasks. Post Rebuttal Comments: This paper is not a theory paper. Also, it shows no practical relevance. After reading the paper, the reviews, and the discussion carefully, I decided to decrease my score for this paper from 4 to 3. My main concerns are: 1) the lack of experiments on downstream tasks such as robustness certification, and 2) the upper bounds of the Lipschitz constant and the corresponding computational times in Table 1 do not clearly show the advantage of the proposed method. The authors did not convince me on these two points in the rebuttal.


Review 4

Summary and Contributions: The manuscript proposed a polynomial optimization formulation to estimate the global and local Lipschitz constant of a multi-layer neural network. The polynomial optimization is then approximated using semidefinite programs based on both dense and sparse Lasserre's Hierarchy. Numerical experiments are performed to show the proposed methods' superior performance compare with LP based relaxations studied in [21]

Strengths: This is a solid work with contributions mostly in the theoretical side. The proposed optimization can be used to numerically estimate the local and global Lipschitz constants (which are important for various neural network applications) for small- to medium- sided networks with ReLU activation.

Weaknesses: The proposed methods and algorithms can only be applied to small networks. There remain tremendous challenges scaling the methods to networks of practical sizes. Most importantly, as the authors commented in the conclusion, the heuristic relation is designed for QCQP that applies to 1-hidden layer networks. (Though extensions are considered in Appendix E). All the numerical simulations are also performed for neural networks with at most one-hidden layer.

Correctness: The claims and method are technically sound and correct to my knowledge.

Clarity: Yes, the paper is well and clearly written.

Relation to Prior Work: Comparison to LP based methods in [21] is extensively performed. Other relevant work are reviewed.

Reproducibility: Yes

Additional Feedback: Some minor comments: - It's good to define the abbreviations of the algorithms shown in Table 1 in the main text. - What is the unit (seconds I guess) for running time reported in Table 1. - Define LBS before it's used in Table 1. - Is HR-1 the same as SHOR? - v in ||v||_* of equation (2) should be bold - Vector-vector multiplications are used in several places, e.g., equation (4). I think they are component wise, but it's good to precisely state them somewhere. - Perhaps include a reference for SDP in footnote 1 on page 4 if you think it's necessary to define SDP for the readers. - Line 132 on page 4, what is y_i? - Define LCEP in the main text when it's first used in line 130 on page 4 (even though you defined it in the section title).


Review 5

Summary and Contributions: [ QUESTION-ONLY NON-REVIEW FROM AC. SCORE IS FAKE. BUT PLEASE ANSWER! ] 1. Can the authors please clarify the theoretical contribution? The exact method appears direct from the definition of gradient, and the heuristic is based on the SoS hierarchy, but oddly, does not carry any approximation guarantees. Is there a hope of an approximation guarantee? 2. Can the authors please clarify the empirical contribution, specifically, a way this could be applied to standard networks? The present demonstration is on 80x80; can the authors predict what happens, including a time estimate, on something like AlexNet, VGGNet, ResNet? If the answer is large, can the others say how the methods already given in the paper are a significant step in lowering these numbers? Does any of the competing work handle these large practical networks?

Strengths: .

Weaknesses: .

Correctness: .

Clarity: .

Relation to Prior Work: .

Reproducibility: Yes

Additional Feedback:

[Author Response · NeurIPS 2020]

We thank all the reviewers for the careful and insightful review of our manuscript. We will of course correct all the minor suggestions and typos provided by all the reviewers. Most reviewers are concerned with handling larger networks and we provide a detailed view on this point at the end, in the response to AC.

**Response to R1:** • Regarding comparison with baselines, we build on reference [21] where detailed comparison with baseline methods were carried out. Accordingly, LipOpt provides superior results compared to concurrent methods. We stick to comparison with LipOpt with these results in mind for simplicity. • We will work on the introduction and moderate the claim about L2 norm emphasizing that this mainly holds for the robustness certification problem.

**Response to R2:** • Regarding scaling for bigger network the reviewer can see the response to AC for a detailed account. • Regarding the validity of the upper bound, the SDP relaxation provides a guaranteed upper bound of the exact optimal value. Thus our heuristic relaxation does give valid upper bounds of the exact Lipschitz constant. • Regarding friendliness of the paper, we provided simple examples for the SDP relaxation in appendix B. We will make more explicit the presence of these examples starting from the introduction.

**Response to R3:** • We are aware of the importance of downstream applications but decided to stick the problem of Lipschitz constant estimation for several reasons: 1/ The main concurrent approach was [21] which did not perform such experiments, we decided to stick to the same evaluation metrics since the methods were designed for it. 2/ Technical and methodological improvements for the problem of Lipschitz constant estimation could possibly be translated to different certification problems. 3/ We stick to the intuition that better estimation of Lipschitz constants will be helpful for downstream applications. • Regarding the advantage of our method, we see in Table 1 that HR-2 actually gets better upper bounds than LipOpt-3 with less running time. Although the results of HR-2 is a little bit worse than LipOpt-4, the running time of LipOpt-4 is much bigger than HR-2. Shor's relaxation presented in the paper is actually also one of our methods. Since we use an exact model to describe the Lipschitz constant, which is different from [21]. The results in Table 2 show that our method (both HR-2 and Shor) works for the MNIST network while LipOpt doesn't. • We also would like to point out that references [1] and [3] proposed by the referee do not consider Lipschitz constants, but rather consider directly the problem of robustness certification. The reference [2] provides an algorithm to compute upper bounds on Lipschitz constants to derive a training procedure in order to obtain a robust network. These (rather coarse) bounds are recursively obtained for each network's component by relying on composition, addition and concatenation rules. We will add completeness experiments on the problem of robustness certification utilizing our method.

**Response to R4:** Indeed the relaxation applied to two layer networks is possible but requires slight modifications. Precisely, SHOR is the first order method for one layer networks, HR-1 is the first order method for two layer networks, and HR-2 stands for the second order method both for one and two layer networks. Pushing further the size of networks which can be considered is one of our middle term goals (see also response to AC).

**Response to AC:** **Theoretical contributions:** • Provide a semi-algebraic representation of the subgradient of the ReLU function. Prove that this can be used to formulate a polynomial optimization problem whose solutions provide certified upper bounds to neural network Lipschitz constants (as noted by one of the reviewers, this is not a direct consequence of the definition of gradients and stands on recent developments in nonsmooth analysis). Derive an adaptation of Lasserre's Hierarchy specifically tailored to the problem of Lipschitz constant estimation of ReLU networks. • Since our model is an exact description of the Lipschitz constant, one obtains convergence (only asymptotic) to the exact constant value with the dense hierarchy. We plan to derive similar guarantees for the proposed sparse hierarchy in future research. Thus the only problem is the scalability. The hierarchy is used here in a non-asymptotic regime, between the first and second orders. Obtaining approximation guarantees for the resulting upper bounds is difficult, most available bounds are very pessimistic, which is coherent with the fact that the hierarchy can solve NP-hard problems. **Empirical contribution:** • We provide results on two hidden layer networks in the appendix (up to size $50 \times 50 \times 10$, the relaxation has to be slightly modified). We run our method on a $784 \times 500$ dense single hidden layer MNIST network (results are reported in the main text). The methods take between 3 and 5 hours to run on a small personal laptop. • Generally speaking, an $L$ hidden layer (fully-connected or convolutional) network results in a polynomial optimization problem whose objective is of degree $L + 1$. And one needs to use the $\lceil (L+1)/2 \rceil$-th order SOS-hierarchy to solve such problems, which contains $O(n^{L+1})$ variables and PSD matrices of size $O(n^{\lceil (L+1)/2 \rceil})$ where $n$ is the total number of variables in the problem. • Considering real networks, such as AlexNet, requires to treat 5 convolution layers and 3 fully-connected layers where in each layer there are more than 4000 nodes. This means that we will have $O(4000^9) \approx 10^{33}$ variables in the targeted optimization problem, which seems impossible. Even if we compute the Lipschitz constant recursively layer by layer, there are still $O(4000^2) \approx 10^7$ variables. Handling problems with million variables requires to handle the very sparse structure provided by these layers in a tailored version of the SOS-hierarchy. This constitutes an interesting venue for future research. To our knowledge, none of the concurrent methods is able, as of today, to handle such real scale networks with similar guarantees (i.e. valid upper bounds and asymptotic convergence). This would require considerable work on the implementation, much more performing hardware and a lot of engineering. • A key in lowering these numbers is to devise relaxation methods specifically tailored to the sparsity patterns of deep neural convolutional networks. Indeed, the associated polynomial optimization problems have a specific sparsity pattern, which can be exploited to significantly reduce the computational burden. This work constitutes a step in this direction and we will continue to push this idea further.

[Meta-Review · NeurIPS 2020]

This paper highlights an issue with computing clarke differentials of shallow ReLU networks, provides a fix, and uses this to design an impractical exact solver and a heuristic for determining Lipschitz constants. As this fix seems relevant to recently accepted papers and quite timely, this is a valuable paper. --- Minor comment. I believe it is a bit opaque to most readers to refer to the conservative vector field stuff. I believe I have a fairly elementary/accessible direct proof of the fact you need? Feel free to include it and say "proof due to anonymous AC"; I have a background in this stuff. Anyway, consider the characterization of the clarke differential as the convex hull of sequences of gradients (this form is equivalent). For Lipschitz constant, we want the maximum norm element, which necessarily happens at a corner of the convex hull, therefore for our purposes it suffices to consider sequences. Since the ReLU network will be almost-everywhere differentiable, we can consider a shrinking sequence of balls around any point, and we will have gradients which are arbitrarily close to any corner of the differential at our given point. Therefore, the norms will converge to that norm, and thus it suffices to optimize over differentiable points, and what we choose at the nondifferentiability does not matter.